# Score-based Source Separation with Applications to Digital Communication Signals

**Tejas Jayashankar**[1]   **Gary C.F. Lee**[1]   **Alejandro Lancho**[1,2]   **Amir Weiss**[1]

**Yury Polyanskiy**[1]   **Gregory W. Wornell**[1]

[1]Massachusetts Institute of Technology   [2]Universidad Carlos III de Madrid

## Abstract

We propose a new method for separating superimposed sources using diffusion-based generative models. Our method relies only on separately trained statistical priors of independent sources to establish a new objective function guided by *maximum a posteriori* estimation with an $\alpha$-*posterior*, across multiple levels of Gaussian smoothing. Motivated by applications in radio-frequency (RF) systems, we are interested in sources with underlying discrete nature and the recovery of encoded bits from a signal of interest, as measured by the bit error rate (BER). Experimental results with RF mixtures demonstrate that our method results in a BER reduction of 95% over classical and existing learning-based methods. Our analysis demonstrates that our proposed method yields solutions that asymptotically approach the modes of an underlying discrete distribution. Furthermore, our method can be viewed as a multi-source extension to the recently proposed score distillation sampling scheme, shedding additional light on its use beyond conditional sampling. The project webpage is available at https://alpha-rgs.github.io.

## 1   Introduction

The problem of single-channel source separation (SCSS) is ubiquitous and arises in many different applications, ranging from the cocktail party problem in the audio domain to interference mitigation in the digital communications domain. Broadly speaking, the goal in SCSS is to decompose a mixture $\mathbf{y} = \kappa_1 \mathbf{x}_1 + \cdots + \kappa_K \mathbf{x}_K \in \mathcal{Y}^N$ into its $K$ constituent components, $\{\mathbf{x}_i\}_{i=1}^K, \mathbf{x}_i \in \mathcal{X}_i^N$. Motivated by applications in modern engineering, e.g., intelligent communication systems, we are particularly interested in the SCSS problem for heterogeneous sources with *underlying discrete structure*.

Prior art that uses deep learning for source separation problems has been well-studied in the audio and image domain by leveraging domain-specific structures. For example, recent end-to-end methods in the audio domain leverage spectrogram masking techniques to separate sparse time-frequency representations [1, 2], while in the visual domain, natural images may be separable by exploiting local features and "smoothness" assumptions in the color space [3, 4]. More recent efforts have tried to solve this problem by leveraging independently trained statistical priors using annealed Langevin dynamics sampling [5–7]. However, these methods have demonstrated shortcomings on discrete sources that exhibit intricate temporal structures with *equiprobable multimodal distributions* [8].

These additional challenges motivate us to develop a *novel general Bayesian framework* for source separation that relies on statistical priors over sources with underlying discreteness. Specifically, we focus on learning data-driven priors for two main reasons—i) **Unknown system parameters**, e.g., the underlying signal generation model may not be available to create hand-crafted priors; and ii) **Automation**, facilitated by learning methods to create plug-and-play priors for versatile use.

37th Conference on Neural Information Processing Systems (NeurIPS 2023).

In this paper, we study the SCSS problem involving a mixture of two signals, $\mathbf{s}$ and $\mathbf{b}$, which interfere at a different relative scale to produce the mixture signal $\mathbf{y} = \mathbf{s} + \kappa \mathbf{b}$, where $\kappa \in \mathbb{R}_+$ is the scaling coefficient. We additionally impose restrictions on access to paired training data samples $(\mathbf{s}, \mathbf{b}, \mathbf{y})$ during training. Our motivation for this restriction arises from the following consideration: given $n$ sources, the number of two-component source separation models leveraging joint-statistics grows as $\mathcal{O}(n^2)$. Any changes to the training data—even for a single source—would require $\mathcal{O}(n)$ model updates. In contrast, solutions that leverage independently trained priors over the sources need to *update only a single model* (i.e., $\mathcal{O}(1)$). Additionally, these priors can also be used to solve more generalized source-separation problems, e.g., with $K > 2$ components or with a different mixture model, *without requiring any additional training*.

We demonstrate the successful operation of our method in the digital radio-frequency (RF) setting. Signals in the RF domain are often discrete in nature as their randomness arises from bits/symbols that modulate a continuous waveform. As such, source separation performance is measured not only through continuous reconstruction metrics, e.g., the mean squared error (MSE), but also (and primarily) by the fidelity of digital communication, commonly measured by the bit error rate (BER).

**Broader Impact.**  The crux of our work is motivated by challenges arising in the next generation of wireless systems. The proliferation of intelligent wireless devices—utilizing finite spectrum resources—has called for better interoperability strategies, so as to ensure reliable operations in a rapidly growing *heterogeneous wireless networking ecosystem* [9–11]. Interference from other sources operating in the same channel, e.g., 5G waveforms or WiFi signals, can lead to deterioration in quality of service. While conventional approaches treat such interference as Gaussian noise, there could be significant performance gains if we were able to learn and exploit intricate statistical structures of these co-channel signals [12]. We take an important step towards this goal by developing data-driven *score-based SCSS solutions* that demonstrate significant gains over modern and conventional approaches, thereby unveiling novel intelligent and effective interference mitigation strategies. We further elaborate on the broader impact of our work in Appendix A.

**Contributions.**  We start from first principles and propose a new Bayesian-inspired method for SCSS. Our main contributions are as follows.

- We present a new method for SCSS that leverages the (approximate) score from pre-trained diffusion models to extend MAP estimation using generalized Bayes' theorem with an $\alpha$-posterior [13, 14] across different levels of Gaussian smoothing. Our method, termed as $\boldsymbol{\alpha}$-**RGS** ($\boldsymbol{\alpha}$-posterior with **R**andomized **G**aussian **S**moothing) (see §3), is easy to implement by randomizing the predetermined training noise levels without cumbersome tuning of a special annealing noise schedule, as in Langevin-dynamics-based approaches [5].

- We show that our formulation is a multi-source extension of the recently proposed score distillation sampling (SDS) objective [15]. Our analysis (see §4) shows that despite randomizing across multiple noise levels, the local extrema of our loss (and hence of SDS) asymptotically approach the modes of the underlying unsmoothened source distribution(s) (see §3.2.1).

- We demonstrate through experimental results (see §5) that $\alpha$-RGS outperforms classical signal processing and annealed Langevin-dynamics-based approaches for RF source separation, with a 96% and 94.5% improvement in BER (averaged across different levels of interference) over traditional and existing learning-based methods, respectively. While score-based methods have been adopted to solve problems such as symbol detection [16] and channel estimation [17], to the best of our knowledge, this is the first work that applies score-based models for the task of source separation in the RF domain.

**Paper Organization.**  The paper is organized as follows. The necessary prerequisites are introduced in §2. §3 meticulously develops our proposed method, starting from basic principles. We provide analytical characterizations of our method in §4, with additional details available in Appendix B. In §5, we introduce our problem in the context of digital communication signals and describe our experiments along with results. A detailed summary of our results is available in Appendix G. A primer on digital communication signals and details about our datasets are available in Appendix C. Details related to diffusion models in the RF context, a primer on conventional baselines, and a description of score-based separation baselines can be found in Appendices D, E and F, respectively. We conclude with §6, where we comment on future work and on the broader impact of our method.

## 2 Prerequisites

### 2.1 Finite Alphabet Signal Processing

In many engineering systems, signals commonly exhibit continuous magnitudes. However, in particular cases, these signals may possess discrete properties, leading to a finite (but possibly large) number of possible realizations. Such signals are often expressed as,

$$s(t) = \sum_{p=-\infty}^{\infty} c_p \, g_p(t - t_p), \tag{1}$$

where $c_p \in \mathcal{S}$ are discrete symbols drawn from a finite set $\mathcal{S} \subset \mathbb{C}$ (or $\mathbb{R}$) and $g_p(\cdot)$ is a continuous filter that "carries" contributions from the symbols. For example, in optical or RF communications, the symbols could correspond to complex-valued mappings of the underlying bits [18], while in the discrete tomography domain, the symbols might correspond to measurements obtained from different materials with a finite number of phases or absorption values [19]. In this work, we will address the challenges associated with finding the global optimum within the optimization landscape formulated for separating a superposition of such discrete sources.

### 2.2 Diffusion Models

Diffusion models are a class of generative models introduced by Sohl-Dickstein et al. [20] based on the principles of thermodynamic diffusion. The model is parametrized by two Markov chains of length $T$—i) the forward chain $q(\mathbf{x}_t | \mathbf{x}_{t-1})$, in which noise is gradually added to the data $\mathbf{x}_0$ such that at the end of the forward process, $q(\mathbf{x}_T) = \mathcal{N}(\mathbf{x}_T; 0, \mathbf{I})$; and ii) the learned reverse chain $p_\phi(\mathbf{x}_{t-1} | \mathbf{x}_t)$, in which noise is gradually removed (i.e., denoising) in order to generate new samples.

These models are trained by minimizing a variational lower bound on the log-likelihood. Recent work by Ho et al. [21] in the image domain shows that this optimization problem translates to learning a denoiser $r_\phi(\mathbf{x}_t, t)$ parametrized as a neural network with weights $\phi$ that estimates the noise term in $\mathbf{x}_t := \sqrt{\alpha_t}\mathbf{x}_0 + \sqrt{1 - \alpha_t}\mathbf{z}$, $\mathbf{z} \sim \mathcal{N}(0, \mathbf{I})$ across different timesteps $t \sim \mathrm{Unif}\left(\{1, \ldots, T\}\right)$,

$$\phi^* = \arg \min_\phi \mathbb{E}_{t,x,z} \left[ \|r_\phi(\mathbf{x}_t, t) - \mathbf{z}\|_2^2 \right], \tag{2}$$

where $\{1 - \alpha_i\}_{i=1}^T$ defines an increasing signal *variance-preserving* noise schedule with $0 \leq \alpha_i \leq 1$. The works in [21–23] show that objective (2) is equivalent to minimizing a loss whose minimizer is an estimate of the marginal score $S_{x_t}(\mathbf{x}_t) = \nabla_{\mathbf{x}_t} \log q(\mathbf{x}_t)$ such that,

$$S_{x_t}(\mathbf{x}_t) \approx -(1 - \alpha_t)^{-1/2} r_\phi(\mathbf{x}_t, t). \tag{3}$$

Other works have extended diffusion models to continuous time using stochastic differential equations [23], while others have used them for signals in the audio domain [24] and forecasting problems [25].

## 3 A Bayesian Framework for Source Separation

We now develop the Bayesian framework underlying our proposed source separation method. We are particularly motivated in developing a generalized framework that relies on statistical priors *only*, without any other form of domain-specific regularization.

### 3.1 *Maximum a Posteriori* (MAP) Source Separation

**Signal statistical structure.** Let $\mathbf{s} \in \mathcal{S} \subset \mathbb{C}^d$ and $\mathbf{b} \in \mathbb{C}^d$ be two *statistically independent* complex-valued vector sources, where $\mathcal{S}$ is a countable set of all realizations $\mathbf{s}$. We assume that $\mathbf{s}$ has PMF $P_\mathbf{s}$, and we let $\mathbf{b}$ be an arbitrary source (potentially discrete with some noise) with PDF $p_\mathbf{b}$. We assume that the latter distributions are multimodal, where the probability is generally (close to) zero except at the modes. Additionally, we seek to be robust to the challenging setting where the distributions have *multiple equiprobable modes*, so as to develop novel methods that can tackle the finite alphabet source separation problem as is motivated in §1. We emphasize that these assumptions do not completely characterize the often complicated fine-grained statistical source structure, and we therefore rely on generative models to learn such unknown structures from data.

**MAP problem formulation.** Consider a mixture composed of two superimposed sources,

$$\mathbf{y} = \mathbf{s} + \kappa \mathbf{b}, \tag{4}$$

where $\kappa \in \mathbb{R}_+$ is a relative scaling coefficient between the two signals.

Given $\mathbf{y}$ and assuming $\kappa$ is known, in order to separate the sources, it is sufficient to estimate $\mathbf{s}$ since $\mathbf{b} = (\mathbf{y} - \mathbf{s})/\kappa$. The MAP estimate of $\mathbf{s}$ is then given by,

$$\widehat{\mathbf{s}} = \arg\max_{\mathbf{s} \in \mathcal{S} \text{ s.t } \mathbf{y} = \mathbf{s} + \kappa \mathbf{b}} p_{\mathsf{s}|\mathsf{y}}(\mathbf{s}|\mathbf{y}). \tag{5}$$

Using Bayes' theorem, (5) can be equivalently expressed as,

$$\arg\max_{\mathbf{s} \in \mathcal{S} \text{ s.t } \mathbf{y} = \mathbf{s} + \kappa \mathbf{b}} p_{\mathsf{s}|\mathsf{y}}(\mathbf{s}|\mathbf{y}) = \arg\max_{\mathbf{s} \in \mathcal{S} \text{ s.t } \mathbf{y} = \mathbf{s} + \kappa \mathbf{b}} p_{\mathsf{y}|\mathsf{s}}(\mathbf{y}|\mathbf{s}) P_{\mathsf{s}}(\mathbf{s}) \tag{6a}$$

$$= \arg\min_{\mathbf{s} \in \mathcal{S}} -\log P_{\mathsf{s}}(\mathbf{s}) - \log p_{\mathsf{b}}\left((\mathbf{y} - \mathbf{s})/\kappa\right), \tag{6b}$$

where the likelihood $p_{\mathsf{y}|\mathsf{s}}(\mathbf{y}|\mathbf{s}) = p_{\mathsf{b}}\left((\mathbf{y} - \mathbf{s})/\kappa\right)$ under the constraint (4), and we convert to negative log probabilities in (6b). Due to the underlying discreteness of $\mathbf{s}$, and the potential underlying discreteness of $\mathbf{b}$ as well, the objective function in (6b) is not differentiable. Hence, in its current form, gradient-based techniques cannot be used to solve this problem, and we must instead resort to combinatorial methods, which are computationally infeasible even for moderate dimension size $d$.

### 3.2 Proposed Method ($\alpha$-RGS)

To overcome the computational complexity of combinatorial-based methods, our goal is to develop new gradient-based SCSS solutions that leverage diffusion models trained on discrete sources. To this end, we propose to use multiple levels of Gaussian smoothing with an extended MAP framework with an $\alpha$-posterior, such that the optimization landscape of the resulting objective function is smoothened.

#### 3.2.1 The Smoothing Model and $\alpha$-posterior Generalized Bayes'

**Surrogate distribution.** One can almost perfectly approximate $P_s$ (in some well-defined sense) with the *surrogate distribution* $p_{\bar{\mathsf{s}}}$, where $\bar{\mathbf{s}} = \mathbf{s} + \varepsilon_s$ for $\varepsilon_s \sim \mathcal{N}(0, \sigma_s^2 \mathbf{I})$, $\sigma_s \to 0$. While now theoretically amenable to optimization via gradient descent, the sharp modes, which constitute numerical pitfalls, often cause gradient-based methods to get stuck in local extrema.

**Gaussian Smoothing Model.** We adopt a *variance-preserving* smoothing model, with adjustable noise levels, based on the formulation of the stochastic process proposed in [21]. Let $\{1 - \alpha_t\}_{t=1}^{T}$ be a sequence of noise levels such that $\alpha_i > \alpha_{i+1}$ and $\alpha_i \in (0, 1]$. Define the "smoothened sources" as

$$\tilde{\mathbf{s}}_t(\bar{\mathbf{s}}) \coloneqq \sqrt{\alpha_t}\bar{\mathbf{s}} + \sqrt{1 - \alpha_t}\mathbf{z}_s, \tag{7a}$$

$$\tilde{\mathbf{b}}_u(\bar{\mathbf{s}}, \mathbf{y}) \coloneqq \sqrt{\alpha_u}\left(\mathbf{y} - \bar{\mathbf{s}}\right)/\kappa + \sqrt{1 - \alpha_u}\mathbf{z}_b, \tag{7b}$$

where $t, u \sim \text{Unif}\left(\{1, \ldots, T\}\right)$ and $\mathbf{z}_s, \mathbf{z}_b \sim \mathcal{N}(0, \mathbf{I}_d)$. Observe that $\tilde{\mathbf{s}}_t$ and $\tilde{\mathbf{b}}_u$—the continuous-valued proxies for s and b, respectively—have PDFs rather than PMFs, and in particular, their PDFs have infinite support, and they are differentiable. More importantly, a direct consequence is that it smoothens the optimization landscape and helps in preventing gradient-based algorithms from getting stuck at spurious local minima.

**Generalized Bayes' with an $\alpha$-posterior.** We found it useful to replace the likelihood in (6a) with a distribution proportional to $p_{\mathsf{y}|\mathsf{s}}(\mathbf{y}|\mathbf{s})^{\omega}$, $\omega > 1$. Intuitively, under the constraint (4), this sharpens the distribution of b and gives a higher weight to the modes of $p_{\mathsf{b}}$ relative to the natural weighting that arises from the MAP criterion. This is beneficial, for example, when $p_{\mathsf{b}}$ is more complicated and has many more modes than $P_s$.

This aforementioned reweighting has been used as an implementation trick in diffusion sampling with classifier conditioning [26], but we recognize this as MAP estimation with an $\alpha$-posterior[1] where $\alpha = \omega$ which is expressed through the generalized Bayes' rule [13, 14, 27, 28] as,

$$\underbrace{p_{\mathsf{s}|\mathsf{y}}(\mathbf{s}|\mathbf{y}; \omega)}_{\alpha\text{-posterior with } \alpha = \omega} \quad \propto \quad \underbrace{p_{\mathsf{y}|\mathsf{s}}(\mathbf{y}|\mathbf{s})^{\omega}}_{\text{tempered likelihood}} \quad \underbrace{P_{\mathsf{s}}(\mathbf{s})}_{\text{prior}}. \tag{8}$$

---

[1] $\alpha$ should not be confused with, and has no relation to, $\alpha_t$ from the Gaussian smoothing model.

---

**Algorithm 1** Proposed Method: $\alpha$-posterior with Randomized Gaussian Smoothing ($\alpha$-RGS)

---

1: **function** SEPARATION($\mathbf{y}, \kappa, N, \{\eta_i\}_{i=0}^{N-1}, \boldsymbol{\theta}^{(0)}$)              $\triangleright$ $N$ total steps, learning rate $\eta_i$ at step $i$
2:      **for** $i \leftarrow 0, N-1$ **do**
3:          $t, u \sim \text{Uni}\{1, \ldots, T\},\ \mathbf{z}_s, \mathbf{z}_b \sim \mathcal{N}(0, \mathbf{I})$              $\triangleright$ Sample random noise levels
4:          $\tilde{\mathbf{s}}_t\left(\boldsymbol{\theta}^{(i)}\right) = \sqrt{\alpha_t}\,\boldsymbol{\theta}^{(i)} + \sqrt{1-\alpha_t}\mathbf{z}_s$              $\triangleright$ Smooth $\mathbf{s}$ at level $t$, (7a)
5:          $\tilde{\mathbf{b}}_u\left(\boldsymbol{\theta}^{(i)}, \mathbf{y}\right) = \sqrt{\alpha_u}\left(\mathbf{y} - \boldsymbol{\theta}^{(i)}\right)/\kappa + \sqrt{1-\alpha_u}\mathbf{z}_b$              $\triangleright$ Smooth $\mathbf{b}$ at level $u$, (7b)
6:          $\widehat{\mathbf{z}}_s, \widehat{\mathbf{z}}_b = r_{\phi_\mathbf{s}}\left(\tilde{\mathbf{s}}_t\left(\boldsymbol{\theta}^{(i)}\right), t\right), r_{\phi_\mathbf{b}}\left(\tilde{\mathbf{b}}_u\left(\boldsymbol{\theta}^{(i)}, \mathbf{y}\right), u\right)$              $\triangleright$ Compute scores, (3)
7:          $\boldsymbol{\theta}^{(i+1)} = \boldsymbol{\theta}^{(i)} - \eta_i\left[\frac{\omega}{\kappa}\frac{\sqrt{\alpha_u}}{\sqrt{1-\alpha_u}}\left(\widehat{\mathbf{z}}_b - \mathbf{z}_b\right) - \frac{\sqrt{\alpha_t}}{\sqrt{1-\alpha_t}}\left(\widehat{\mathbf{z}}_s - \mathbf{z}_s\right)\right]$              $\triangleright$ Subtract noise, (12)
8:      **end for**
9:      **return** $\widehat{\mathbf{s}} = \boldsymbol{\theta}^{(N)}$, $\widehat{\mathbf{b}} = (\mathbf{y} - \widehat{\mathbf{s}})/\kappa$
10: **end function**

---

In practice, using an $\alpha$-posterior has demonstrated increased learning speeds [13, 14] and could potentially help in resolving model mismatch arising from the use of approximate densities or scores (rather than exact ones) during optimization via gradient descent.

**Single noise level estimation loss.**    Let $\widehat{\mathbf{s}}(\boldsymbol{\theta}) = \boldsymbol{\theta}$ be our estimate of $\mathbf{s}$.[2] Motivated by the form in (6b), we generalize using the smoothing (7a)-(7b) in conjunction with the $\alpha$-posterior with $\alpha = \omega$ to define a new *single noise level estimation loss*,

$$\mathcal{L}_{t,u}(\boldsymbol{\theta}) := -\log p_{\tilde{\mathbf{s}}_t}\left(\tilde{\mathbf{s}}_t\left(\boldsymbol{\theta}\right)\right) - \omega \log p_{\tilde{\mathbf{b}}_u}\left(\tilde{\mathbf{b}}_u\left(\boldsymbol{\theta}, \mathbf{y}\right)\right). \tag{9}$$

While departing from the original MAP (6b), the newly-defined approximated loss (9) thereof facilitates gradient-based methods, which is key to our solution approach. Particularly, by varying the level of smoothing, (9) is more easily explored in regions between the modes via gradient descent.

### 3.2.2   Estimation Rule Across Multiple Noise Levels

Intuitively, larger noise variances allow us to move between modes during gradient descent. In contrast, at lower noise levels the modes are sharper, which is beneficial in resolving the solution, assuming the iterative procedure starts at the basin of attraction of the correct local extremum point, or, alternatively, another "good" local extremum point. We propose a new estimation rule that uses (9) across multiple noise levels. Randomizing over multiple levels of Gaussian smoothing, our proposed gradient update asymptotically (in the number of iterations) takes the form,

$$\nabla_{\boldsymbol{\theta}}\mathcal{L}(\boldsymbol{\theta}) := \underbrace{-\mathbb{E}_{\mathsf{t},\mathbf{z}_t}\left[\sqrt{\alpha_t}\,S_{\tilde{\mathbf{s}}_t}\left(\tilde{\mathbf{s}}_t\left(\boldsymbol{\theta}\right)\right)\right] + \frac{\omega}{\kappa}\mathbb{E}_{\mathsf{u},\mathbf{z}_u}\left[\sqrt{\alpha_u}S_{\tilde{\mathbf{b}}_u}\left(\tilde{\mathbf{b}}_u\left(\boldsymbol{\theta}, \mathbf{y}\right)\right)\right]}_{\mathbb{E}_{\mathsf{t},u}[\nabla_{\boldsymbol{\theta}}\mathcal{L}_{t,u}(\boldsymbol{\theta})]}, \tag{10}$$

where $t, u \sim \text{Unif}\left(\{1, \ldots, T\}\right)$, and the true score is defined as

$$S_{\tilde{\mathbf{s}}_t}\left(\tilde{\mathbf{s}}_t\left(\boldsymbol{\theta}\right)\right) := \nabla_{\mathbf{x}}\log p_{\tilde{\mathbf{s}}_t}\left(\mathbf{x}\right)\Big|_{\mathbf{x}=\tilde{\mathbf{s}}_t(\boldsymbol{\theta})}, \tag{11}$$

and similarly for $S_{\tilde{\mathbf{b}}_u}$.

Our proposed updates can be implemented using stochastic gradient descent as shown in Algorithm 1. We use *pre-trained diffusion models* (§2.2) to approximate the score when deriving the analytical score is not possible. Using (3), we relate the learned score to the denoiser available from the diffusion model, and also use a zero mean noise corrected estimate,

$$\mathbb{E}_{\mathbf{z}_s}\left[r_{\phi_\mathbf{s}}\left(\tilde{\mathbf{s}}_t\left(\boldsymbol{\theta}\right), t\right)\right] = \mathbb{E}_{\mathbf{z}_s}\left[r_{\phi_\mathbf{s}}\left(\tilde{\mathbf{s}}_t\left(\boldsymbol{\theta}\right), t\right) - \mathbf{z}_s\right], \tag{12}$$

in our updates as shown in Algorithm 1. The above rule is motivated to introduce numerical stability and reduces the variance of the updates, since the diffusion models were trained to minimize the squared value of the same error term in (2). The state-of-the-art method of score distillation sampling (SDS) [15] that uses diffusion models as critics, also uses the *same term* in their gradient update step.

---

[2]Owing to the continuous nature of the optimization landscape, $\boldsymbol{\theta}$ is in fact an estimate of $\bar{\mathbf{s}}$ (see §3.2.1), and does not affect BER and MSE estimates significantly.

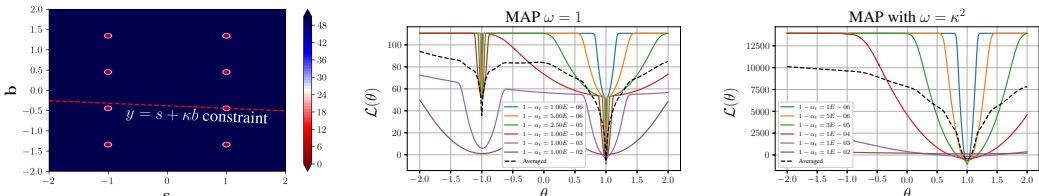

Figure 1: **Left:** Two discrete sources, with infinitesimal additive noise, superimposed to produce a joint distribution with 8 equiprobable modes. An observed mixture $\mathbf{y}$, imposes a linear constraint in this space. **Middle:** Extending vanilla MAP ($\omega = 1$) to multiple noise levels still has a relatively large local minima. **Right:** By using $\omega = \kappa^2$, we are able to accentuate the correct mode and smooth the landscape even further. Colored curves correspond to (15) evaluated with $T = 1$ and $t = u$.

### 3.3 Relation to Other Works

**BASIS separation.** Jayaram and Thickstun [5] introduced the BASIS separation algorithm that leverages the score from a generative model to perform source separation using annealed Langevin dynamics. Unlike our method, the BASIS algorithm relies on a specially tuned noise schedule for separation that is distinct from the diffusion model training noise schedule. We circumvent the challenges associated with tuning such a schedule and instead *re-use the pre-determined training noise schedule* in a randomized fashion. As such, *the only parameter that we tune is the learning rate*. Our results (§5.3) show that we outperform BASIS in the context of RF source separation.

**Score Distillation Sampling.** Our work can be viewed as an extension of the recently proposed Dreamfusion architecture [15], which uses pre-trained image diffusion models as critics to guide the generation of novel 3D visual content. Given a generated sample, $g(\boldsymbol{\theta})$, Score Distillation Sampling (SDS) updates the 3D object realization using gradient descent with a gradient given by,

$$\nabla_{\theta}\mathcal{L}_{\text{SDS}}\left(\phi, \tilde{\mathbf{s}} = g(\boldsymbol{\theta})\right) = \mathbb{E}_{\mathsf{t}, \mathsf{z}}\left[w(t)(r_{\phi}(\sqrt{\alpha_t}\,g(\boldsymbol{\theta}) + \sqrt{1 - \alpha_t}\mathbf{z}, t) - \mathbf{z})\right], \tag{13}$$

where $w(t)$ is a scaling related to the noise variance. Our updates contain the same gradient terms in (13) and hence it can be viewed as a *multi-source extension of SDS*, shedding light on its use in applications to problems beyond sampling with interactions between numerous individual priors.

**Diffusion models as priors.** More recently diffusion priors have been used in inverse problems such as MRI reconstruction [29, 30], image restoration/colorization [23, 31–34] and for developing general inversion methods [32, 35]. A majority of these works solve a MAP problem with domain-specific likelihood constraints, thus resembling our formulation. However, they either use annealed Langevin dynamics or a noise contracting reverse diffusion process to sample from the posterior distribution, necessitating the tuning of a special annealing schedule. In our work, we demonstrate convergence to the same modes using a simpler scheme based on randomized levels of Gaussian smoothing (see §4). The $\alpha$-posterior parameter can be interpreted as a weighting of the regularization term, thus shedding light on the potential applicability of our method to general inverse problems in the future.

## 4 Characterization of $\alpha$-RGS

In this section, we characterize the behavior of our objective function through a simple yet intuitive example. We first present a sufficient condition for perfect signal separation in the context of discrete sources that follow the signal generation model in (1).

**Proposition 1.** *Let $s(t)$ and $b(t)$ be two sources following (1) with underlying symbols $c_p^s$ and $c_p^b$ respectively. Assume that the symbols are obtained as,*

$$c_p^s = f\left(\{u_i\}_{i=p}^{L+p}\right) \quad and \quad c_p^b = h\left(\{v_i\}_{i=p}^{L+p}\right), \tag{14}$$

*where $f : \mathcal{U}^L \to \mathbb{C}$ and $h : \mathcal{V}^L \to \mathbb{C}$ are mappings from a sequence of length $L$ over the discrete alphabets $\mathcal{U}$ and $\mathcal{V}$ respectively[3]. If the mapping between the discrete representation and the symbol representation of the sources is unique, perfect recovery is possible.*

---

[3]In digital communications, the discrete alphabet is typically the binary alphabet, representing the underlying bits that need to be communicated, and the symbols refer to the digital constellation (see §5.1)

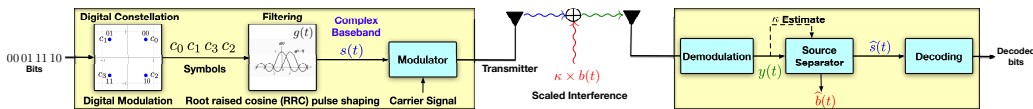

Figure 2: The single-carrier digital modulation pipeline with an intelligent decoder that performs a pre-processing stage of source separation. Illustrated is an example with a QPSK constellation and a root-raised cosine (RRC) pulse shaping function.

We now argue that under such conditions, our method in Algorithm 1 asymptotically approaches a local extremum corresponding to the following loss function,

$$\mathcal{L}(\boldsymbol{\theta}) \coloneqq -\mathbb{E}_{\mathsf{t},\mathsf{z}_s} \left[ \log p_{\tilde{\mathsf{s}}_t} \left( \tilde{\mathbf{s}}_t \left( \boldsymbol{\theta} \right) \right) \right] - \omega \mathbb{E}_{\mathsf{u},\mathsf{z}_u} \left[ \log p_{\tilde{\mathsf{b}}_u} \left( \tilde{\mathbf{b}}_u \left( \boldsymbol{\theta}, \mathbf{y} \right) \right) \right]. \tag{15}$$

To see this, let s and b be two discrete sources with equiprobable modes at $\{-1, +1\}$ and $\{\pm 6/\sqrt{20}, \pm 2/\sqrt{20}\}$ respectively. Figure 1 shows the contours of the negative log joint probability, with the sources augmented with an infinitesimally small amount of Gaussian noise (see §3.2.1). An observed mixture $y$, adds a linear constraint in the $(s, b)$ plane. The goal of Algorithm 1 is to pick out the constraint-satisfying mode at $(1, -2/\sqrt{20})$. If $\omega = 1$, as shown in the middle plot, and given a poor initialization $\boldsymbol{\theta}^{(0)}$ closer to $-1$, the gradients in this region may prevent the estimate from escaping the suboptimal local minimum. If instead, we use $\omega = \kappa^2 > 1$ ($\kappa = 15.85$), as shown in rightmost plot, the optimization landscape is better conditioned in the same (absolute) neighborhood around $\boldsymbol{\theta} = -1$ with the mode at $+1$ much more accentuated. Specifically, if the algorithm is now initialized at $\boldsymbol{\theta}^{(0)} = -1$, after enough iterations, gradient steps at larger noise levels would lead the solution towards the direction of $\boldsymbol{\theta} = 1$. Thus, on average (black curve), after enough iterations, the solution will approach $+1$. In contrast, Langevin-dynamics-based approaches without the $\alpha$-posterior weighting [5], if initialized poorly, could potentially get stuck at local extrema due to sharpening of the optimizing landscape as the level of noise decreases.

The estimate of the discrete source s can be obtained by mapping the solution returned by Algorithm 1 to the closest point (in the Euclidean sense) in the discrete alphabet $\mathcal{S}$. This relies on the extremum $\boldsymbol{\theta}^*$ of (15) being sufficiently close to the desired mode of $p_{\mathsf{s}|\mathsf{y}}(\mathbf{s}|\mathbf{y}; \omega)$, so that $\boldsymbol{\theta}^*$ can be mapped to the correct point in $\mathcal{S}$ with high probability. In the context of the above example, a key observation in achieving this desired behavior is that the mode at $(1, -2/\sqrt{20})$ is still prominent at large noise levels. Thus, randomizing across different noise levels helps balance the exploration between modes and the resolution of the estimate. This mainly requires a suitable noise level range (i.e., a lower bound on $\alpha_T$), to ensure that the modes are sufficiently resolved. In our experiments, we show that *no additional tuning is required* and that the training noise levels from the pre-trained diffusion models can be re-used, provided that the models have learned the source's structure sufficiently well.

A more detailed characterization of our algorithm's convergence to the modes can be found in Appendix B. We underscore that in this work we focus on general signal types and mixtures, where perfect separability is not guaranteed and hence performance bounds are not analytically tractable.

## 5    Applications and Experiments

### 5.1    Digital Communication Signals

**Digital communication pipeline.**    At a high level, digital communications deals with the transmission of bits by modulating a so-called "carrier signal". Groups of bits, from which the underlying discreteness of these sources originates, are first mapped to symbols $c_p \in \mathbb{C}$ via the *digital constellation*—a mapping between groups of bits and a finite set of complex-valued symbols. These symbols are subsequently aggregated via (1) to form a *complex-valued continuous waveform*. In the RF domain, $g(\cdot)$ is known as the *pulse shaping* filter and helps limit the signal's bandwidth [36, Sec 4.4.3]. The constellation is chosen (among other considerations) by the number of bits modulated simultaneously. Common schemes include modulating two bits at a time (Quadrature Phase Shift Keying, or QPSK), or one bit at a time (Binary Phase Shift Keying, or BPSK). Figure 2 illustrates a representative modulation pipeline. To recover the bits at the receiver, one may adopt *matched filtering* (MF) [18, Sec 5.8] before the estimation of the underlying symbols, and thereafter decode them back to bits. For commonly used pulse shaping functions, such as the root-raised cosine

(RRC) shown in Figure 2, the matched filter and pulse shaping filter coincide. We refer readers to [18, 36, 37] for a more thorough exposition of the topic.

**Interference mitigation.**    Mitigating co-channel interference is a challenging problem, especially in heterogeneous networks [9–11]. If the system parameters and signal generation model are known ahead of time, one can leverage this knowledge to devise hand-crafted priors. However, one often deals with interference from more complicated wireless sources, for which the signal model is unknown. In such scenarios, data-driven methods that learn priors from background recordings can be useful. Our results demonstrate that $\alpha$-RGS, which leverages these priors, sets a new state-of-the-art in SCSS for RF mixtures. We envision that our SCSS solution could help mitigate such interference prior to decoding the signal, as shown in the right hand (receiver) side of Figure 2.

**Deep learning for RF systems.**    Recently deep learning methods have demonstrated the potential to reap significant gains over handcrafted model-based methods in RF applications [38, 39]. Some works have studied the problem of symbol detection [40, 41], where they assume that the channel is stationary. Other works, such as DeepSIC [42], use deep learning for interference cancellation in the multi-user setting within the same channel. In contrast, we deal with the more challenging setting of non-stationary interference, thereby requiring efficient exploitation of intricate temporal structures. While the latter works consider the superposition of independent and identically distributed sources (same technology), we assume unknown additive interference (cross technology), a hard problem to solve with naïve decoding methods in the absence of explicit prior knowledge about the interference. Our problem formulation is closer to recent work in [43]. However, they learn an end-to-end estimate of the signal from paired data samples. As motivated in §1, we instead assume restricted access to joint data, with a focus on capturing properties of the components through independent priors.

## 5.2   Experimental Setup

We now detail the setup for training diffusion models on RF signals. We subsequently explain how these models are used in the implementation of our method at inference time (i.e, for separation).

**RF SCSS formulation.**    We are interested in recovering $\mathbf{s}$, the signal of interest (SOI), from a mixture $\mathbf{y} = \mathbf{s} + \kappa\mathbf{b}$, where $\mathbf{b}$ is assumed to be a co-channel interference with unknown system parameters. We evaluate performance using two metrics—i) the MSE, that measures the distortion between the estimated SOI and the ground truth; and ii) the BER of the decoded discrete representation, which is obtained from the estimated SOI by extracting the underlying bits (see Appendix C). The latter measure is particular to digital communication signals as it captures the fidelity of the estimated representation that is only partially reflected in the MSE criterion.

**Datasets.**    We train diffusion models on different RF datasets —i) synthetic QPSK signals with RRC pulse shaping (see §5.1), ii) synthetic OFDM signals (BPSK and QPSK) with structure similar to IEEE 802.11 WiFi signals; and iii) signals corresponding to "CommSignal2" from the RF Challenge [44], which contains datasets of over-the-air recorded signals. All synthetic datasets were created using the NVIDIA Sionna toolkit [45]. All datasets contain between 150k - 500k samples and we use a 90-10 train-validation split during training. Additional details are in Appendix C.

**Diffusion model training.**    We adopt the Diffwave [24] architecture for our experiments, with a minor changes (see Appendix D) to accommodate the complex-valued nature of our signals. Our models are trained in the waveform domain on inputs of length 2560 with the real and imaginary components concatenated in the channel dimension. We train unconditional diffusion models and assume no access to knowledge about the signal generation model. We use noise variance levels in the range $(1\mathrm{e}{-}4, 0.72)$ discretized into 50 levels. We train for 500k steps with early stopping on 2 x NVIDIA 3090 GPUs. Detailed hyperparameters for our training setup are provided in Appendix D.

**Source separation setup.**    We consider three different mixtures in our experiments, all using an RRC-QPSK signal as the SOI $\mathbf{s}$. The interference signal $\mathbf{b}$ is one of OFDM (BPSK), OFDM (QPSK) or a windowed recording from the CommSignal2 dataset. Our proposed algorithm uses $\omega = \kappa^2$ and is initialized with the MF solution given the mixture $\mathbf{y}$ (see §5.1). Note that $\kappa$ can be equivalently described as the signal to interference ratio (SIR $:= 1/\kappa^2 := \mathbb{E}_{\mathsf{s}}\left[\|\mathbf{s}\|_2^2\right]/\mathbb{E}_{\mathsf{b}}\left[\|\kappa\mathbf{b}\|_2^2\right]$).

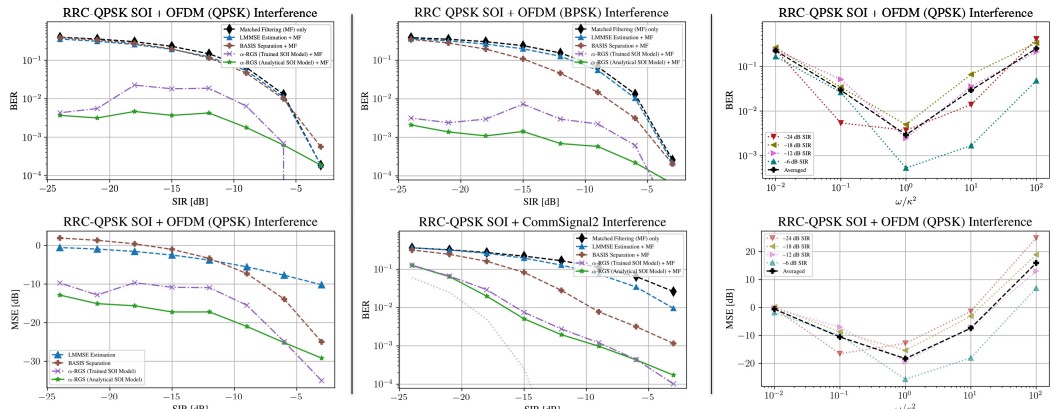

Figure 3: **Left:** Source separation results for a mixture with RRC-QPSK SOI and OFDM (QPSK) interference. All curves are obtained by averaging 400 different mixtures per SIR and using $\omega = \kappa^2$. Our proposed method significantly outperforms traditional and learning-based baselines (BASIS) in terms of BER and MSE across all noise levels. **Middle:** Similar comparisons only in BER for mixtures with OFDM (BPSK) and CommSignal2 interference, respectively. BER is slightly higher for the CommSignal2 mixture since it contains background noise that is amplified for large $\kappa$. The **black dotted line** in the bottom figure denotes the (presumed) BER lower bound assuming the background noise is an additive white Gaussian noise. **Right:** BER and MSE versus $\omega/\kappa^2$ for different SIR levels. Clearly a good choice in this setting, in the sense of minimum BER and MSE, is $\omega = \kappa^2$.

We assume that $\kappa$ is known[4] and use $N = 20,000$ with a cosine annealing learning rate schedule [46]. The OFDM mixtures use $(\eta_{\max}, \eta_{\min}) = (5e{-}3, 1e{-}6)$ and the CommSignal2 mixture uses $(\eta_{\max}, \eta_{\min}) = (2e{-}3, 1e{-}6)$. Importantly, we *re-use the training noise levels* from the diffusion models and randomize over all but the smallest noise level resulting in a noise variance range of $(1 - \alpha_1 = 1.2e{-}3, 1 - \alpha_T = 0.72)$ discretized into $T = 49$ levels. We test performance across SIR levels ranging from $-24$ dB to $-3$ dB ("strong interference" regime), by changing the value of $\kappa$ in the mixture. Each set of separation experiments was conducted on a single NVIDIA V100 GPU.

**Baselines.** We compare our proposed method against baselines that also leverage independent statistical or structural priors over the sources. The simplest baseline, which nevertheless is still commonly used in most communication systems, is the matched filtering solution, which treats the interference as white Gaussian noise. The linear minimum mean square error (LMMSE) solution, a commonly used technique for signal estimation, is another baseline that leverages (up to) second-order statistics of the underlying source distributions. More details on these methods are in Appendix E.

We also compare with the BASIS separation algorithm (see §3.3), which is the closest learning-based method that resembles our problem formulation. Applying their method as is yielded poor results, and hence we modified the original hyperparamters by tuning the annealing schedule to the best of our abilities for a fair comparison. For more details on this, refer to Appendix F.

To study the fidelity of our learned score models, we derive the analytical score function of the QPSK SOI in the symbol domain (i.e., before pulse shaping). We use this analytical score as another comparison to demonstrate the performance of our method if the score was known perfectly. This formulation is closer to a learning-based interference mitigation setting, where we assume perfect knowledge about the SOI model, and rely on a learned interference model.

### 5.3 Source Separation Results

Figure 3 shows the source separation results for the three different mixture types in §5.2, obtained by averaging 100 independent trials per SIR. Our proposed method (analytical or trained SOI score) significantly outperforms MF, LMMSE and BASIS in terms of both BER and MSE. As expected, our best results are obtained by leveraging the prior knowledge, in the form of the analytical score

---

[4]Many communication systems have power constraints and equalization capabilities, and with the endowment of such knowledge it is possible to estimate the signal to interference ratio (SIR) within reasonable margin.

for the SOI. Nevertheless, our learned SOI score can nearly mimic this performance, and despite the slight degradation still outperforms all baselines in terms of BER as well. It should be noted that CommSignal2 contains small amount of background noise, which is amplified at low SIR (high $\kappa$). This noise constrains the minimal achievable BER even under the assumption of only having the residual AWGN present, illustrated by the black dotted line at the bottom middle plot in Figure 3. We also outperform the baselines in terms of MSE across all SIR levels, an example of which is shown at the bottom left of Figure 3. More detailed results are included in Appendix G.

**Characterizing the choice of** $\omega$**.** We numerically verify that $\omega = \kappa^2$ is a good choice for the $\alpha$-posterior term. To this end, we find a suitable order of magnitude for $\omega$, by varying $\omega/\kappa^2$ between $10^{-2}$ and $10^2$ and computing the resulting BER and MSE for four representative SIR levels using a held-out *hyperparameter validation set* of 400 mixtures. The right panel of Figure 3 shows that $\omega = \kappa^2$ achieves the minimum BER and MSE on average. More characterizations are in Appendix G.

## 6 Concluding Remarks

In this work, we propose $\alpha$-RGS, a method that extends MAP estimation with an $\alpha$-posterior across randomized levels of Gaussian smoothing, which stems from a new objective function, whose extrema points correspond to the modes of the underlying discrete distribution of interest. Our method relies only on pre-trained diffusion models as priors via a simple randomized algorithm that does not require cumbersome tuning of a special annealing schedule, as is done in existing Langevin-dynamics-based works. Through simple analytical illustrations, we demonstrate the favorable mode-preserving nature of our objective, and show that $\alpha$-RGS is a generalized extension of the recently proposed SDS loss. Experiments on RF sources demonstrate superior separation performance, with gains of up to 95% in terms of both BER and MSE over classical and existing score-based SCSS methods.

**Limitations and future work.** We believe there is a potential to use our algorithm to develop a general toolkit of algorithms for source separation and inverse problems. However, we do list some *limitations* of our method that will be a focus of future work: 1) inference time ($\sim$ 5 min per mixture) requires speed-up in real-time systems; 2) studying settings with more than two mixture components or other mixture models; 3) developing a systematic approach to choose $\omega$ for practitioners; and 4) addressing the robustness of our method for applications to signals with new properties, e.g., gravitational waves. We further elaborate on the broader impact of our work in Appendix A.

## Acknowledgments and Disclosure of Funding

Research was supported, in part, by the United States Air Force Research Laboratory and the United States Air Force Artificial Intelligence Accelerator under Cooperative Agreement Number FA8750-19-2-1000. The views and conclusions contained in this document are those of the authors and should not be interpreted as representing the official policies, either expressed or implied, of the United States Air Force or the U.S. Government. The U.S. Government is authorized to reproduce and distribute reprints for Government purposes notwithstanding any copyright notation herein.

The authors acknowledge the MIT SuperCloud and Lincoln Laboratory Supercomputing Center for providing HPC resources that have contributed to the research results reported within this paper.

Alejandro Lancho has received funding from the European Union's Horizon 2020 Research and Innovation Programme under the Marie Sklodowska-Curie Grant Agreement No. 101024432. This work is also supported, in part, by the National Science Foundation (NSF) under Grant No. CCF-2131115.

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

# Supplementary Material

## A    Broader Impact

### A.1    Regarding the Proposed $\alpha$-RGS Method

**Contribution and Distinction from Existing Works.**   Our method is in the effort of i) using score-based/diffusion-based methods beyond sampling, and to solve inverse problems, and ii) developing source separation strategies with a focus on sources with underlying discrete properties, which may make it differ from existing works where the usual goal is to obtain perceptually close solutions, which is not a good criterion in the RF domain.

**Scalable Approach to Source Separation with Pre-Trained Models.**   An attractive property of our method is that by using pre-trained models as independent priors, we minimize the need for expensive compute resources to retrain models to tackle different scenarios/different mixture settings.

**Trade-off in Computation Time.**   While the training time is reduced, the inference computation time is as of now longer than other methods (e.g., end-to-end training with mixtures as inputs and SOIs as desired outputs), thus making it unsuitable for real-time operation. As we discuss about its potential use for applications, speeding up the algorithmic aspects of the operation is essential in furthering this method towards practical applications.

**Ethical Considerations of our Work.**   The intersection between generative AI and RF signals is a nascent field and hence, their ethical implications have been less explored. Nevertheless, we believe that this is an area that requires special attention to address issues such as malicious parties learning and spoofing, among others. While these are not the focus and the intent of our work, we believe that works in security in this space are actively exploring such problems.

**Privacy Considerations with Generative Models.**   Where privacy and sensitive information is concerned, our work solely focuses on synthetic and public datasets. No sensitive transmissions have been captured, and therefore learned, by our generative models.

### A.2    Regarding Applications to RF Systems

**Novelty, Relevance and Timeliness.**   To the best of our knowledge, this is the first work that uses score-based methods for physical layer transmission in digital communication systems. The interference mitigation problem has significant relevance to enabling reliable operations in next-generation wireless systems. With the growth of heterogeneous wireless networks, there is overcrowding and growing scarcity of wireless spectrum. This work explores interference mitigation solutions with the goal of enabling efficient sharing of the finite resources to allow growth of the wireless ecosystem in a sustainable way.

**Proof-of-Concept Exploration.**   Our work demonstrates via a proof-of-concept what might be possible for non-orthogonal access[5] to explore opportunities for new generation of transmission strategies and communication protocols; particularly, the emphasis on the low SIR regime is in line with motivations in lower-power device operations.

**Considerations arising from Demonstration via Computational Simulations.**   All our experiments are done via computational simulation and thus, we did not operate in RF bands that we do not have rights to access. Our synthetic simulations may differ from practice where unmodeled physical phenomenon may occur, thus necessitating natural extensions of our work to deal with such impediments.

---

[5]That is, neighboring communicating devices that operate of the same, shared physical resources (e.g., time and frequency).

# B Additional Characterizations of $\alpha$-RGS

We now "dissect" (15) in §4, and focus on a single term,

$$\mathcal{L}_{\mathbf{s}}(\boldsymbol{\theta}) := -\mathbb{E}_{\mathbf{t},\mathbf{z}_s}\left[\log p_{\tilde{\mathbf{s}}_t}\left(\tilde{\mathbf{s}}_t\left(\boldsymbol{\theta}\right)\right)\right], \tag{16}$$

where we recall the definition of the Gaussian smoothing model from §3.2.1,

$$\tilde{\mathbf{s}}_t\left(\boldsymbol{\theta}\right) = \sqrt{\alpha_t}\bar{\boldsymbol{\theta}} + \sqrt{1-\alpha_t}\mathbf{z}_s, \quad \mathbf{z}_s \sim \mathcal{N}(0,\mathbf{I}_d). \tag{17}$$

As detailed in §3.2.1, when estimating a discrete source $\mathbf{s}$, $\bar{\boldsymbol{\theta}} = \boldsymbol{\theta} + \varepsilon_s$ for $\varepsilon \sim \mathcal{N}(0,\sigma_s^2\mathbf{I})$, $\sigma_s \to 0$, is a continuous surrogate of the discrete estimate $\boldsymbol{\theta}$, useful for optimization via gradient descent. The above loss only depends on the prior of the source, and is in particular independent of the data. Therefore, this term serves (and can be viewed) as a regularizer in our inference optimization problem. Although analysis of (16) is (strictly speaking) not useful in order to make statements about (15), it is nevertheless informative and insightful to show that the local extrema of (16) approach the modes of the underlying source distribution $P_{\mathbf{s}}$, by solving for the stationary points,

$$\mathbb{E}_{\mathbf{t},\mathbf{z}_s}\left[\nabla_{\boldsymbol{\theta}}\log p_{\tilde{\mathbf{s}}_t}(\gamma_t\boldsymbol{\theta} + \sigma_t\mathbf{z}_s)\right] = 0 \tag{18}$$

where we have used Leibniz rule for differentiation under the integral and where we have defined the shorthand notation,

$$\gamma_t := \sqrt{\alpha_t} \quad \text{and} \quad \sigma_t := \sqrt{1-\alpha_t}. \tag{19}$$

Through our examples we will demonstrate the mode-approaching nature of the solutions to (18) thus establishing another interpretation of the asymptotic behavior of our method—two loss terms whose stationary points are modes of the corresponding source distributions which work together in unison to satisfy the constraints imposed by the observed mixture. We thereby end up, with high probability, in the correct mass-points of the underlying distributions.

## B.1 Multivariate Normal Sources

We first start with the analysis of a multivariate normal source and show that the the *exact* mode is obtained as the local extremum of (16).

**Proposition 2.** *Let $\mathbf{s} \in \mathbb{R}^d$ be a multivariate normal source with mean $\boldsymbol{\mu}_{\mathbf{s}}$ and covariance matrix $\boldsymbol{\Sigma}_{\mathbf{s}}$. For the Gaussian smoothing model in (17), the score at timestep $t$ is*

$$\nabla_{(\tilde{\mathbf{s}}_t(\mathbf{s}))}\log p_{\tilde{\mathbf{s}}_t}\left(\tilde{\mathbf{s}}_t\left(\mathbf{s}\right)\right) = -\boldsymbol{\Sigma}_{\tilde{\mathbf{s}}_t}^{-1}\left(\gamma_t\mathbf{s} + \sigma_t\mathbf{z}_t - \boldsymbol{\mu}_{\tilde{\mathbf{s}}_t}\right), \tag{20}$$

$$\boldsymbol{\mu}_{\tilde{\mathbf{s}}_t} := \gamma_t\boldsymbol{\mu}_{\mathbf{s}} \quad \text{and} \quad \boldsymbol{\Sigma}_{\tilde{\mathbf{s}}_t} := \gamma_t^2\boldsymbol{\Sigma}_{\mathbf{s}} + \sigma_t^2\mathbf{I}.$$

*Then, the minimizer $\boldsymbol{\theta}^* = \arg\min_{\boldsymbol{\theta}}\mathcal{L}_{\mathbf{s}}(\boldsymbol{\theta})$ is equal to $\boldsymbol{\mu}_{\mathbf{s}}$, i.e., the mode of the source distribution.*

*Proof.* Since $\mathbf{s}$ is a continuous source, we perform optimization over a continuous space with respect to $\boldsymbol{\theta} \in \mathbb{R}^d$.[6] Notice that,

$$\nabla_{\boldsymbol{\theta}}\log p_{\tilde{\mathbf{s}}_t}(\gamma_t\boldsymbol{\theta} + \sigma_t\mathbf{z}_s) = \gamma_t\,\nabla_{(\tilde{\mathbf{s}}_t(\boldsymbol{\theta}))}\log p_{\tilde{\mathbf{s}}_t}\left(\tilde{\mathbf{s}}_t\left(\boldsymbol{\theta}\right)\right) \tag{21a}$$

$$= \gamma_t\boldsymbol{\Sigma}_{\tilde{\mathbf{s}}_t}^{-1}\left(\gamma_t\boldsymbol{\theta} + \sigma_t\mathbf{z}_s - \boldsymbol{\mu}_{\tilde{\mathbf{s}}_t}\right) \tag{21b}$$

Substituting (21b) in (18), we see that the minimizer must satisfy

$$\mathbb{E}_{\mathbf{t}}\left[\boldsymbol{\Sigma}_{\tilde{\mathbf{s}}_t}^{-1}\left(\gamma_t\boldsymbol{\theta} - \boldsymbol{\mu}_{\tilde{\mathbf{s}}_t}\right)\right] = 0, \tag{22}$$

where we have used the fact that $\mathbf{z}_s$ is a zero mean Gaussian realization. Thus, the minimizer is

$$\boldsymbol{\theta}^* = \left(\sum_{t=1}^T\gamma_t\boldsymbol{\Sigma}_{\tilde{\mathbf{s}}_t}^{-1}\right)^{-1}\left(\sum_{t=1}^T\boldsymbol{\Sigma}_{\tilde{\mathbf{s}}_t}^{-1}\boldsymbol{\mu}_{\tilde{\mathbf{s}}_t}\right), \tag{23}$$

which can be simplified further as

$$\boldsymbol{\theta}^* = \left(\sum_{t=1}^T\gamma_t\boldsymbol{\Sigma}_{\tilde{\mathbf{s}}_t}^{-1}\right)^{-1}\left(\sum_{t=1}^T\gamma_t\boldsymbol{\Sigma}_{\tilde{\mathbf{s}}_t}^{-1}\right)\boldsymbol{\mu}_{\mathbf{s}} = \boldsymbol{\mu}_{\mathbf{s}}, \tag{24}$$

where we have used (2) to substitute $\boldsymbol{\mu}_{\tilde{\mathbf{s}}_t} = \gamma_t\boldsymbol{\mu}_{\mathbf{s}}$. $\qquad\square$

Thus, we have established that the local extremum of the (16) corresponds to the mode in the multivariate Gaussian case.

---

[6]In this continuous setting, a surrogate distribution is not needed and $\bar{\boldsymbol{\theta}} = \boldsymbol{\theta}$ in (17)

## B.2 Tweedie's Formula and Digital RF Sources

We next analyze (16) in the context of RF source separation by studying sources defined by a digital constellation with symbols from the finite set $\mathcal{A}$. Before proceeding, we first state Tweedie's formula [47], a tool that will aid us in computing the scores of arbitrary distributions. For completeness of the exposition, we prove it for the case of scalar random variables below.

**Lemma 1.** *(Tweedie's formula) Let* $\mathsf{a}$ *be a random variable with an arbitrary distribution and let* $\mathsf{z}_\sigma$ *be a normal random variable with mean* $\mu$ *and variance* $\sigma^2$ *independent of* $\mathsf{a}$. *The score of the random variable* $\mathsf{x} := \mathsf{a} + \mathsf{z}_\sigma$ *is*

$$\nabla_x \log p_{\mathsf{x}}(x) = \frac{\mu - x}{\sigma^2} + \frac{1}{\sigma^2} \mathbb{E}_{\mathsf{a}|\mathsf{x}}\left[a | x = a + z_\sigma\right]. \tag{25}$$

*Proof.* Since $\mathsf{x}$ is a sum of two independent random variable, it follows a convolution distribution given by,

$$p_{\mathsf{x}}(x) = (p_{\mathsf{a}} * p_{\mathsf{z}_\sigma})(x). \tag{26}$$

Therefore,

$$\nabla_x \log(p_{\mathsf{a}} * p_{\mathsf{z}_\sigma})(x) = \frac{1}{(p_{\mathsf{a}} * p_{\mathsf{z}_\sigma})(x)} (p_{\mathsf{a}} * \nabla_x p_{\mathsf{z}_\sigma})(x) \tag{27}$$

$$= \frac{1}{(p_{\mathsf{a}} * p_{\mathsf{z}_\sigma})(x)} \int p_{\mathsf{a}}(t) \nabla_x p_{\mathsf{z}_\sigma}(x - t)\,\mathrm{d}t \tag{28}$$

$$= \frac{1}{(p_{\mathsf{a}} * p_{\mathsf{z}_\sigma})(x)} \int p_{\mathsf{a}}(t) p_{\mathsf{z}_\sigma}(x - t) \frac{(\mu - x + t)}{\sigma^2}\,\mathrm{d}t \tag{29}$$

$$= \frac{\mu - x}{\sigma^2} + \frac{1}{\sigma^2} \int t p_{\mathsf{a}|\mathsf{a}+\mathsf{z}_\sigma}(t|x)\,\mathrm{d}t \tag{30}$$

$$= \frac{\mu - x}{\sigma^2} + \frac{1}{\sigma^2} \mathbb{E}[a | x = a + z_\sigma]. \tag{31}$$

where (29) follows from,

$$\nabla_x p_{\mathsf{z}_\sigma}(x - t) = -p_{\mathsf{z}_\sigma}(x - t) \frac{(\mu - x + t)}{\sigma^2}. \tag{32}$$

$\square$

**Proposition 3.** *Let* $\mathsf{s}$ *be a scalar random variable representing a symbol drawn uniformly from the finite constellation set* $\mathcal{A} \subset \mathbb{C}$. *For the Gaussian smoothing model in* (17) *where, at timestep* $t$,

$$\tilde{s}_t(s) = \gamma_t s + \sigma_t z_t,$$

*the score is*

$$\nabla_{(\tilde{s}_t(s))} \log p_{\tilde{\mathsf{s}}_t}(\tilde{s}_t(s)) = \frac{1}{\sigma_t^2}\left(-\tilde{s}_t(s) + \sum_{a \in \mathcal{A}} a \cdot \phi_t(a, \tilde{s}_t(s))\right), \tag{33}$$

*where,*

$$\phi_t(a, \tilde{s}_t(s)) := \frac{\exp\{-\frac{|\tilde{s}_t(s) - \gamma_t a|^2}{\sigma_t^2}\}}{\sum_{\bar{a} \in \mathcal{A}} \exp\{-\frac{|\tilde{s}_t(s) - \gamma_t \bar{a}|^2}{\sigma_t^2}\}}. \tag{34}$$

*Proof.* We will use Lemma 1 to compute the score, for which the only quantity we need to compute is the conditional expectation. Let $\mathsf{s}' := \gamma_t \mathsf{s}$. Since $\mathsf{s}$ is discrete, $P_{\mathsf{s}'}$ is also a uniform distribution over

the symbol set $\mathcal{A}_t := \{\gamma_t a \mid a \in \mathcal{A}\}$. By Bayes' rule,

$$\mathbb{P}\left[\mathbf{s}' = \gamma_t a | \mathbf{s}' + \sigma_t \mathbf{z}_s = x\right] = \frac{\mathbb{P}\left[\mathbf{s}' + \sigma_t \mathbf{z}_s = x | \mathbf{s}' = \gamma_t a\right] \mathbb{P}\left[\mathbf{s}' = \gamma_t a\right]}{\sum_{\bar{a} \in \mathcal{A}} \mathbb{P}\left[\mathbf{s}' + \sigma_t \mathbf{z}_s = x | \mathbf{s}' = \gamma_t \bar{a}\right] \mathbb{P}\left[\mathbf{s}' = \gamma_t \bar{a}\right]} \tag{35}$$

$$= \frac{\mathbb{P}\left[\mathbf{z}_s = \frac{x - \gamma_t a}{\sigma_t}\right] \cdot \frac{1}{|\mathcal{A}|}}{\sum_{\bar{a} \in \mathcal{A}} \mathbb{P}\left[\mathbf{z}_s = \frac{x - \gamma_t \bar{a}}{\sigma_t}\right] \cdot \frac{1}{|\mathcal{A}|}} \tag{36}$$

$$= \frac{\exp\{-\frac{|x - \gamma_t a|^2}{\sigma_t^2}\}}{\sum_{\bar{a} \in \mathcal{A}} \exp\{-\frac{|x - \gamma_t \bar{a}|^2}{\sigma_t^2}\}} \tag{37}$$

$$= \phi_t(a, x). \tag{38}$$

Therefore, from Lemma 1,

$$\nabla_{(\tilde{s}_t(s))} \log p_{\tilde{\mathbf{s}}_t}(\tilde{s}_t(s)) = \frac{1}{\sigma_t^2}\left(-\tilde{s}_t(s) + \sum_{a \in \mathcal{A}} a \cdot \phi_t(a, \tilde{s}_t(s))\right)$$

$$\square$$

As an illustrative example, consider a BPSK source with modes at $-1$ and $+1$. To solve (18) for $\theta$, we require the score of the BPSK source at different non-zero levels of Gaussian smoothing. Note that since the source is discrete, the score is undefined if no smoothing is applied. Though this smoothened score can be computed using (33), the stationary points of (18) cannot be computed in closed-form. Hence, in order to study the behavior of the solutions, we simulate (16), as shown in Figure 4. The colored curves plot the loss for a single noise level, i.e., when $T = 1$, while the black curve computes the loss across multiple levels of smoothing, by computing an average over the single noise level curves. It is evident from these curves, that while the minima might not lie at the modes $-1$ and $+1$ for the single noise level curves, by averaging across multiple noise levels, the solution(s) to (18) approach the mode(s) of the source distribution. As pointed out in §4, an important caveat

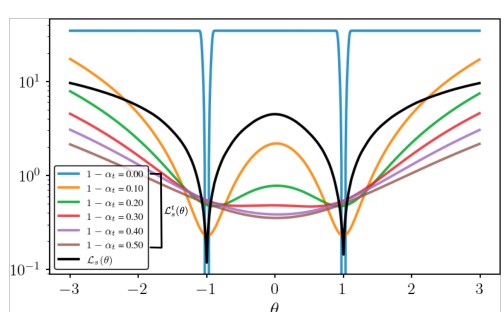

Figure 4: Simulating (16) on a BPSK source. The loss at individual timestamps is visualized in addition to the total loss. The minima are at the modes of the BPSK source distribution, $-1$ and $+1$. Larger noise levels allow for exploration between modes and smaller noise levels sharpen the mode-seeking behavior.

is that this behavior is only observed for a good choice of the noise level range.[7] Larger noise levels aid in moving between the modes and smaller noise levels help resolving the solution. Only using larger noise levels, for example, can smoothen out the landscape to the point that the modes are no longer discernible, thus resulting in erroneous estimates. We leverage the noise levels from pre-trained diffusion models that have learned the statistical structure sufficiently well, thus avoiding cumbersome tuning of the noise level range.

### B.3  Gaussian Mixture Source Model

Finally, we consider a scalar Gaussian mixture model (GMM) that can often model arbitrary complicated scalar distributions. We derive the score for a 2-component GMM and then generalize the result to an arbitrary $K$-component GMM.

**Proposition 4.** *Consider a two-component scalar Gaussian mixture source,*

$$p_{\mathbf{s}}(s) = \lambda \mathcal{N}(s; \mu_1, \sigma_1^2) + (1 - \lambda) \mathcal{N}(s; \mu_2, \sigma_2^2), \quad \lambda \in (0, 1). \tag{39}$$

---

[7]This only corresponds to choosing an appropriate lower bound on $1 - \alpha_T$ and should not be confused with tuning a noise schedule.

*Under the Gaussian smoothing model in* (17),

$$\nabla_{\tilde{s}_t(s)} \log p_{\tilde{s}_t}(\tilde{s}_t(s)) = \frac{1}{Z(\tilde{s}_t(s))} \left[ \lambda\, c_1(\tilde{s}_t(s)) \left( \frac{\gamma_t \mu_1 - \tilde{s}_t(s)}{\gamma_t^2 \sigma_1^2 + \sigma_t^2} \right) \right.$$
$$\left. + (1 - \lambda)\, c_2(\tilde{s}_t(s)) \left( \frac{\gamma_t \mu_2 - \tilde{s}_t(s)}{\gamma_t^2 \sigma_2^2 + \sigma_t^2} \right) \right], \qquad (40)$$

*where*

$$c_1(\tilde{s}_t(s)) := \mathcal{N}(\tilde{s}_t(s); \gamma_t \mu_1, \gamma_t^2 \sigma_1^2 + \sigma_t^2), \qquad (41)$$
$$c_2(\tilde{s}_t(s)) := \mathcal{N}(\tilde{s}_t(s); \gamma_t \mu_2, \gamma_t^2 \sigma_2^2 + \sigma_t^2), \qquad (42)$$
$$Z(\tilde{s}_t(s)) := \lambda\, c_1(\tilde{s}_t(s)) + (1 - \lambda)\, c_2(\tilde{s}_t(s)). \qquad (43)$$

*Proof.* We need to compute the score of the following distribution,

$$p_{\tilde{s}_t}(\tilde{s}_t(s)) = p_{\gamma_t s + \sigma_t z}(\gamma_t s + \sigma_t z_s = \tilde{s}_t(s)), \qquad (44)$$

Our goal, to this end, is to leverage Lemma 1 to compute the score, thereby requiring an expression for the conditional expectation in (31). Since $s$ and $z_s$ are independent, (44) can be written as a convolution between two distributions,

$$p_{\tilde{s}_t}(\tilde{s}_t(s)) = (p_{s'} * p_{z_{\sigma_t}})(s' + z_{\sigma_t} = \tilde{s}_t(s)), \qquad (45)$$

where

$$s' := \gamma_t s \quad \text{and} \quad z_{\sigma_t} := \sigma_t z_s. \qquad (46)$$

In order to compute the conditional expectation, we first show that,

$$p_{s'+z_{\sigma_t}|s'}(s' + z_{\sigma_t} = \tilde{s}_t(s)|s' = a') p_{s'}(s' = a')$$
$$= \frac{1}{Z(\tilde{s}_t(s))} \left[ \lambda\, c_1(\tilde{s}_t(s)) \mathcal{N}\left( a; \frac{\gamma_t^2 \sigma_1^2 \tilde{s}_t(s) + \gamma_t \sigma_t^2 \mu_1}{\gamma_t^2 \sigma_1^2 + \sigma_t^2}, \frac{\gamma_t^2 \sigma_t^2 \sigma_1^2}{\gamma_t^2 \sigma_1^2 + \sigma_t^2} \right) \right.$$
$$\left. + (1 - \lambda)\, c_2(\tilde{s}_t(s)) \mathcal{N}\left( a; \frac{\gamma_t^2 \sigma_2^2 \tilde{s}_t(s) + \gamma_t \sigma_t^2 \mu_2}{\gamma_t^2 \sigma_2^2 + \sigma_t^2}, \frac{\gamma_t^2 \sigma_t^2 \sigma_2^2}{\gamma_t^2 \sigma_2^2 + \sigma_t^2} \right) \right], \qquad (47)$$

where

$$c_1(\tilde{s}_t(s)) := \mathcal{N}(\tilde{s}_t(s); \gamma_t \mu_1, \gamma_t^2 \sigma_1^2 + \sigma_t^2),$$
$$c_2(\tilde{s}_t(s)) := \mathcal{N}(\tilde{s}_t(s); \gamma_t \mu_2, \gamma_t^2 \sigma_2^2 + \sigma_t^2),$$
$$Z(\tilde{s}_t(s)) := \lambda\, c_1(\tilde{s}_t(s)) + (1 - \lambda)\, c_2(\tilde{s}_t(s)).$$

We start by deriving the density functions for the random variables in (46). Since the convolution between two Gaussians is a Gaussian,

$$p_{s'}(a) = \lambda \mathcal{N}(a; \gamma_t \mu_1, \gamma_t^2 \sigma_1^2) + (1 - \lambda) \mathcal{N}(a; \gamma_t \mu_2, \gamma_t^2 \sigma_2^2), \qquad (48)$$
$$p_{z_{\sigma_t}}(w) = \mathcal{N}(w; 0, \sigma_t^2), \qquad (49)$$

from which it follows, for example through the identities derived in [48, Sec 1]), that

$$p_{\tilde{s}_t}(\tilde{s}_t(s)) = \lambda \mathcal{N}(\tilde{s}_t(s); \gamma_t \mu_1, \gamma_t^2 \sigma_1^2 + \sigma_t^2) + (1 - \lambda) \mathcal{N}(\tilde{s}_t(s); \gamma_t \mu_2, \gamma_t^2 \sigma_2^2 + \sigma_t^2). \qquad (50)$$

We conclude the derivation of the conditional distribution by using Bayes' rule,

$$p_{s'|s'+z_{\sigma_t}}(s' = a|s' + z_{\sigma_t} = \tilde{s}_t(s)) = \frac{p_{s'+z_{\sigma_t}|s'}(s' + z_{\sigma_t} = \tilde{s}_t(s)|s' = a) p_{s'}(s' = a)}{\int p_{s'+z_{\sigma_t}|s'}(s' + z_{\sigma_t} = \tilde{s}_t(s)|s' = a') p_{s'}(s' = a') \, \mathrm{d}a'}. \qquad (51)$$

whereby simplifying the terms in the numerator further, we have

$$p_{s'+z_{\sigma_t}|s'}(s' + z_{\sigma_t} = \tilde{s}_t(s)|s' = a) = p_{z_{\sigma_t}}(z_{\sigma_t} = \tilde{s}_t(s) - a) = \mathcal{N}(a; \tilde{s}_t(s), \sigma_t^2). \qquad (52)$$

Finally, by following the Gaussian product identities in [48, Sec 1], between the distribution in (52) and (48) we reach the result in (47).

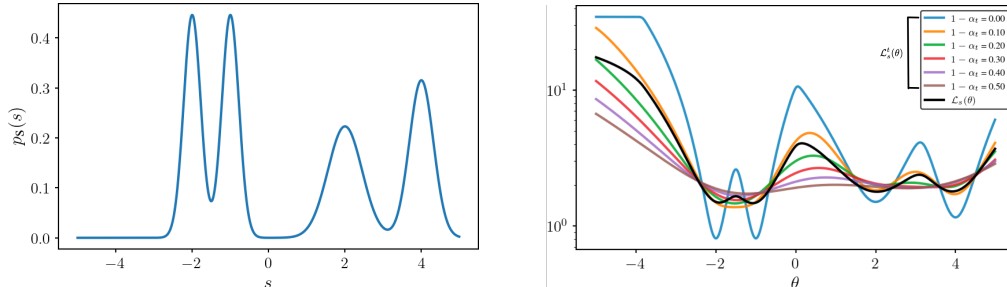

Figure 5: **Left:** A GMM source with two equiprobable modes at $-1$ and $-2$. Two smaller modes are present at $+2$ and $+4$. **Right:** The minima of (16) lie (approximately) at the modes of the source distribution (black curve). Colored curves correspond to (16) evaluated with $T = 1$.

Now we can compute the expectation as,

$$\mathbb{E}\left[\mathsf{s}'|\mathsf{s}' + \mathsf{z}_{\sigma_t} = \tilde{s}_t(s)\right] = \frac{1}{Z(\tilde{s}_t(s))}\left[\lambda\, c_1(\tilde{s}_t(s))\left(\frac{\gamma_t^2\sigma_1^2\tilde{s}_t(s) + \gamma_t\sigma_t^2\mu_1}{\gamma_t^2\sigma_1^2 + \sigma_t^2}\right)\right.$$
$$\left. + (1-\lambda)\, c_2(\tilde{s}_t(s))\left(\frac{\gamma_t^2\sigma_2^2\tilde{s}_t(s) + \gamma_t\sigma_t^2\mu_2}{\gamma_t^2\sigma_2^2 + \sigma_t^2}\right)\right]. \quad (53)$$

Using Lemma 1 we can express the score as,

$$\nabla_{\tilde{s}_t(s)} \log p_{\tilde{\mathsf{s}}_t}(\tilde{s}_t(s)) = \frac{1}{Z(\tilde{s}_t(s))}\left[\lambda\, c_1(\tilde{s}_t(s))\left(\frac{\gamma_t\mu_1 - \tilde{s}_t(s)}{\gamma_t^2\sigma_1^2 + \sigma_t^2}\right)\right.$$
$$\left. + (1-\lambda)\, c_2(\tilde{s}_t(s))\left(\frac{\gamma_t\mu_2 - \tilde{s}_t(s)}{\gamma_t^2\sigma_2^2 + \sigma_t^2}\right)\right].$$

$\square$

We can generalize the score to an arbitrary $K$-component GMM by exploiting symmetries in (40).

**Proposition 5.** *Consider a $K$-component Gaussian mixture source,*

$$p_{\mathsf{s}}(s) = \sum_{i=1}^{K}\lambda_i\,\mathcal{N}(s; \mu_i, \sigma_i^2), \quad \sum_{i=1}^{K}\lambda_i = 1. \quad (54)$$

*For the Gaussian smoothing model in (17), the score at timestep $t$ is,*

$$\nabla_{\tilde{s}_t(s)} \log p_{\tilde{\mathsf{s}}_t}(\tilde{s}_t(s)) = \frac{1}{Z(\tilde{s}_t(s))}\sum_{i=1}^{K}\lambda_i\, c_i(\tilde{s}_t(s))\left(\frac{\gamma_t\mu_i - \tilde{s}_t(s)}{\gamma_t^2\sigma_i^2 + \sigma_t^2}\right), \quad (55)$$

*where*

$$c_i(\tilde{s}_t(s)) \coloneqq \mathcal{N}(\tilde{s}_t(s); \gamma_t\mu_i, \gamma_t^2\sigma_i^2 + \sigma_t^2), \quad (56)$$

$$Z(\tilde{s}_t(s)) \coloneqq \sum_{i=1}^{K}\lambda_i\, c_i(\tilde{s}_t(s)). \quad (57)$$

Similar to the BPSK case, since a closed-form expression for the solution to (18) cannot be obtained, we simulate (16) for a four-component GMM with two large equiprobable modes and two smaller modes. As shown in the colored curves on the right in Figure 5, we again notice similar behavior where the modes are sharper at lower noise levels, with the sharpness decreasing at large noise levels. Hence, gradient-based methods can leverage the randomly chosen larger noise levels to move between modes and use the smaller noise levels to resolve the estimate. On average, the minima of the loss approach the modes of the original GMM source, as shown by the black curve on the right in Figure 5.

## C  Datasets

In this section, we particularize the signal models introduced in §2.1 to RF signals. We first consider a single-carrier signal modulating, e.g., QPSK symbols, which can be expressed as,

$$s(t) = \sum_{p=-\infty}^{\infty} c_{p,\ell} \, g(t - pT_s). \tag{58}$$

The second form corresponds to multi-carrier signals, such as OFDM signals, where multiple symbols are modulated in parallel,

$$b(t) = \sum_{p=-\infty}^{\infty} \sum_{\ell=0}^{L-1} c_{p,\ell} \, \exp\{j2\pi\ell t/L\}. \tag{59}$$

Both types of signals can be grouped in a single family of signals represented as,

$$u(t) = \sum_{p=-\infty}^{\infty} \sum_{\ell=0}^{L-1} c_{p,\ell} \, g(t - pT_s, \ell) \, \exp\{j2\pi\ell t/L\}. \tag{60}$$

We use $s(t)$ to represent the signal of interest (SOI) and $\mathbf{s}$ as the vector representation for a collection of $N$ consecutive samples thereof. Similarly, the interference is represented by $b(t)$ or $\mathbf{b}$.

### C.1  Terminology

Before we describe the construction of our datasets, we will provide some terminology that we use in our description. This terminology is meant to supplement the details already provided in §5.1 in the main paper.

**Symbol constellation:**  A digital constellation describes the underlying scheme for mapping bits to symbols. Typically used digital constellations (the mapping between bits and symbols) include binary phase shift keying (BPSK) and quadrature phase shift keying (QPSK). Specific BPSK and QPSK examples are depicted in Figure 6.

**Pulse/Symbol shaping function:**  This corresponds to $g(\cdot, \cdot)$ in (60) and is meant to help limit the bandwidth of the signal. Each symbol $c_{p,l}$ is multiplied with a pulse shaping function with different time offsets and a main lobe width $T_s$, helping smoothen the signal and remove high frequencies. In this work, we will make use of the root-raised cosine (RRC) pulse shaping for our single-carrier QPSK SOI. On the other had, OFDM signals do not use an explicit pulse shaping function (as shown in (59)), i.e., $g(t, \cdot)$ corresponds to a rectangular function in (60).

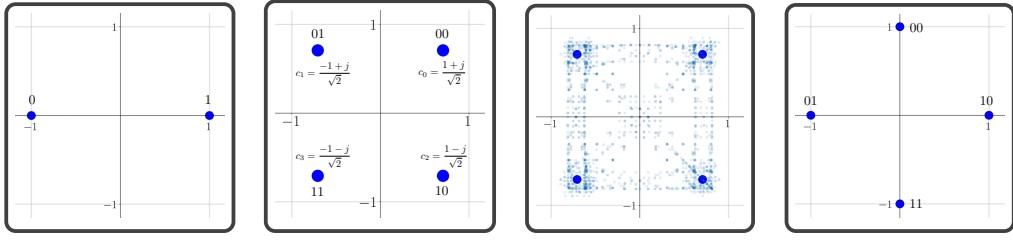

(a) BPSK constellation.    (b) QPSK constellation.    (c) Effect of RRC filtering.    (d) Effect of $45°$ phase.

Figure 6: Discrete constellations for a BPSK and QPSK signal. Application of the RRC filter in the time domain leads to interpolation between constellation point as shown in (c). Phase offsets in the waveform domain can be corrected by looking at the rotated constellation as in (d).

**Total number of subcarriers:**  This corresponds to $L$ in (60) and specifically to OFDM signals of the form in (59). The subcarriers are complex exponentials of the form $\exp\{j2\pi\ell/L\}, \ell \in \{0, \ldots, L-1\}$ with spacing $2\pi/L$ in the (angular) frequency domain, and are orthogonal to each

other. Note that the weighted summation with complex exponentials can be interpreted as computing the *inverse discrete Fourier transform* (IDFT) of the pulse-shaped symbols, for which the inverse fast Fourier transform (IFFT) is used during implementation. Thus, in an interference-free setting, decoding the symbols from $b(t)$ can be performed by computing the FFT of $b(t)$.

**Starting point:** When decoding an RF waveform to extract the underlying symbols, we need them to be *time-synchronized* to a specified frame of reference known as the starting point, $\tau$. If synchronized correctly, this indicates the number of time samples to stream in before bit decoding can begin. If unsynchronized, this corresponds to a shift in time that results in "non-natural" looking decoded constellations at all other time shifts except for the correct time-shift $\tau$. Here, decoding of $u(t)$ can be performed either via an FFT for OFDM, or sampling after matched filtering for a single-carrier signal.

### C.2 Synthetic and Over-the-air RF Datasets

Below we describe the details relating to the RF signals used in our single-channel source separation (SCSS) experiments. To simulate representative types of RF signals, we generated single-carrier QPSK signals with RRC pulse shaping and OFDM signals using the NVIDIA Sionna library [45]. We also used over-the-air recordings corresponding to "CommSignal2" from the dataset in the RF challenge. We do not have explicit knowledge on CommSignal2 source model as it is not provided in the challenge. Hence, we are unable to explicitly describe the pulse shaping function of such a signal type—leaving learning of such properties to our data-driven diffusion models instead.

**RRC-QPSK.** We use QPSK signals with root raised cosine (RRC) pulse shaping [18, Sec 11.3] as the SOI across all our experiments. These signals are used in a variety of different protocols ranging from low-power radios to cellular signals and even for satellite communication. For our experiments, we generated a dataset of 100,000 RRC-QPSK waveforms with the following parameters:

- **Modulation scheme**: QPSK (2 bits per symbol)
- **Pulse shaping function**: RRC, roll-off factor 0.5, spanning $\sim$8 symbols
- **Default starting point**: $\tau_s = 8$
- **QPSK-specific details**:
  - **Oversampling rate**: 16 [samples/symbol], using zero-padding between symbols.

**OFDM (QPSK/BPSK).** For the OFDM signals (specifically, cyclic prefix OFDM), we generated waveforms with the following parameters (so as to simulate an 802.11n WiFi waveform):

- **Modulation scheme**: Either QPSK (2 bits per symbol) or BPSK (1 bit per symbol)
- **Total number of subcarriers (FFT size)**: 64
- **Symbol shaping**: None as per (59) or rectangular window as per (60)
- **Other notes**: Random time offset and random phase rotation for each time series realization
- **OFDM-specific details**:
  - **Number of non-zero subcarriers**: 56 (null at DC subcarrier ($\ell = 0$))
  - **Length of cyclic prefix**: 16 (25% of FFT size)
  - **Number of non-zero symbols**: 56 symbols per 80 samples (1792 symbols in a 2,560 window)
  - **Symbol length**: 80 (64 subcarriers + cyclic prefix of length 16)

In our experiments, we choose to use OFDM signals as one class of interference signals. This means that, given a new realization, we assume that the starting point of a new OFDM symbol (i.e., $\tau_b$) is *unknown*. Further, we assume no access to knowledge about the parameters outlined above, such as the number of subcarriers, the FFT size and the length of cyclic prefix. Hence, we seek a data-driven approach where characteristics of this signal should be learned through data.

Table 1: Hyperparameters used for training our diffusion models.

|  | QPSK | OFDM (BPSK) | OFDM (QPSK) | CommSignal2 |
|---|---|---|---|---|
| Number of residual blocks ($R$) | 30 | 30 | 30 | 30 |
| Dilation cycle length ($m$) | 10 | 10 | 10 | 10 |
| Residual channels ($C$) | 64 | 128 | 256 | 128 |
| Random shift + phase rotation | ✗ | ✓ | ✓ | ✓ |
| Batch size | 128 | 128 | 64 | 128 |
| Learning rate | 5e−4 | 5e−4 | 5e−4 | 1e−4 |
| Early stopping iteration | 360,000 | 220,000 | 340,000 | 90,000 |

**CommSignal2.** CommSignal2 is a dataset comprising over-the-air recordings of type of communication waveforms, provided in [44]. This dataset contains 150 frames each of length 43,560 samples. For purposes of this work, we use the first 100 frames in creating our training set for training the diffusion model, and we extract windows of relevant lengths (2560 samples) from it, to create a dataset of size $\sim 160,000$ samples. The last 50 frames are reserved for creating our test set. Similar to the OFDM case, we also perform random time offsets and phase rotations[8] for each time series realization.

## D  Diffusion Models for RF Signals

We train diffusion models to learn the statistical structure inherent to digital RF signals. As detailed in Appendix C, these signals exhibit both continuous and discrete properties. To this end, we adopt an architecture based on an open-sourced implementation[9] of the DiffWave [24] diffusion model, initially developed for speech synthesis.

Broadly speaking, as shown in [24, Figure 2, Appendix C], DiffWave employs $R$ residual blocks with dilated convolutions, where the output of block $i-1$ serves as an input to block $i$, $i \in \{0, \ldots, R-1\}$. The dilated convolutions assist in the learning of long range temporal and periodic structures. The dilations first start small and successively get larger, such that the dilation at block $i$ is given by $2^{i \bmod m}$, where $m$ is the dilation cycle length. For example, if the dilation periodicity is $m = 10$, then in block $i = 9$ the dilation is 512 and in block 10 the dilation is reset to 1. This allows the network to efficiently tradeoff between learning local and global temporal structures. All residual blocks use the same number of channels, $C$.

**Modifications.** Several changes have been made in order to facilitate training with RF signals. First, since we are dealing with complex-valued continuous waveforms, we train on two channel signals where the real and imaginary components of the RF signals are concatenated in the channel dimension. Second, while the open-sourced implementation uses an $\ell_1$ loss for training, we train with an MSE (squared $\ell_2$) loss to match the training objective in [21]. We monitor the validation MSE loss, and once the loss stops decreasing substantially, we choose to stop training early. Lastly, we had to increase the channel dimension $C$ to learn more complicated RF signals, e.g., OFDM (QPSK). We train all our models on $2 \times$ NVIDIA 3090 GPUs. Additionally, during data loading, we perform random time shifts and phase rotations on the OFDM (BPSK), OFDM (QPSK) and CommSignal2 signals. Physically, these simulate transmission impairments in RF systems. The hyperparameters that we use for training each model are shown in Table 1.

**Metrics.** In the image domain, metrics such as the Fréchet Inception Distance (FID) [49] can be used to assess the quality of generative models. However, such perceptual metrics are less relevant to the digital communications domain. In the RF domain, the fidelity is measured by the ability to extract the underlying transmitted bits. Hence, in the context of our synthetic datasets, we probe generated samples and compare the estimated received symbols with the underlying constellation. However, for real world signals such as CommSignal2, for which we do not know the system parameters, we have no other means to assess the fidelity apart from looking at time-domain structure. This is another

---

[8]Namely, multiplication of the signal with a complex exponential with a random (imaginary) exponent.
[9]https://github.com/lmnt-com/diffwave

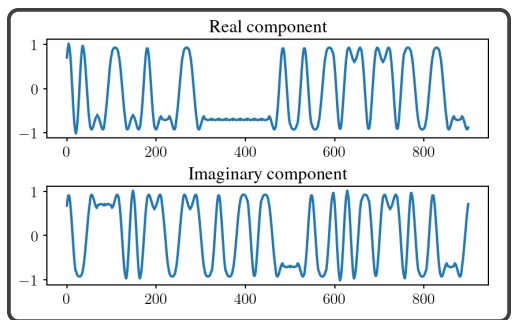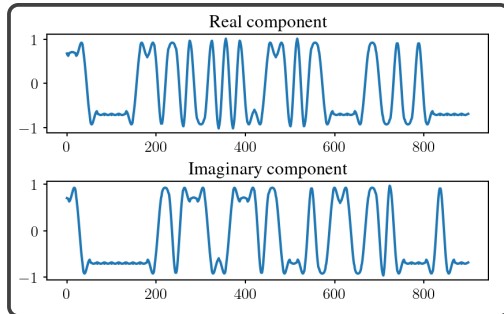

Figure 7: **Left:** A ground truth RRC-QPSK time-domain waveform. **Right:** A sample generated by our trained RRC-QPSK diffusion model. Evidently, the generated waveform resembles the true one.

motivation for studying RF source separation, as it provides us with a framework to assess the quality of the learned statistical priors.

## D.1 RRC-QPSK

We train a diffusion model on the RRC-QPSK dataset (see Appendix C.2). While a QPSK signal has four distinct constellation points, application of the RRC filter results in interpolation between points as shown in Figure 6c. Figure 7 shows an example of a ground truth RRC-QPSK waveform on the left. On the right we show a RRC-QPSK sample generated by our diffusion model. While it is hard to judge whether it has learned the underlying discrete structure, visually, the waveform seems to have characteristics similar to the ground truth.

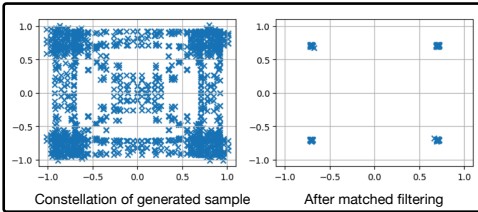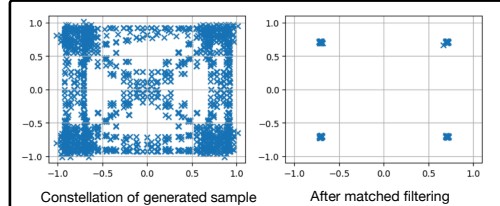

Figure 8: Two samples generated from the diffusion model trained on RRC-QPSK samples. Within each box, the image on the left is underlying constellation of the realization generated by the diffusion model. Notice that the RRC filtering results in interpolation between the QPSK constellation points. After applying MF, the effects of pulse shaping are reversed and the original QPSK constellation is recovered.

In Figure 8, we show that by probing generated samples for the underlying (interpolated) symbols, we do indeed recover the constellation for an RRC-QPSK signal. Note that we can do such probing since we know the parameters and signal generation model for an RRC-QPSK signal. Furthermore, by performing MF and removing the effects of pulse shaping we are able to recover the original QPSK constellation, thus demonstrating that the diffusion model has indeed successfully learned the discrete constellation along with the pulse shaping function from data samples.

## D.2 OFDM (BPSK and QPSK)

We train two OFDM diffusion models, on the OFDM (BPSK) and the OFDM (QPSK) dataset respectively (see Appendix C.2). Figure 9, shows an example of an OFDM (BPSK) signal from our dataset. On the left we plot the real and imaginary components of the waveform. These components visually look like Gaussian noise and it is not evident that there is actually inherent structure to these signals. As mentioned in Appendix C.2, the OFDM signals can be demodulated using oracle knowledge about the FFT size and the cyclic prefix. Additionally, since the signal undergoes a random time shift and phase rotation, the underlying (rotated) BPSK constellation will only be visible when the time shift has been compensated for, i.e., the signal has been time-synchronized, as shown on the right in Figure 9.

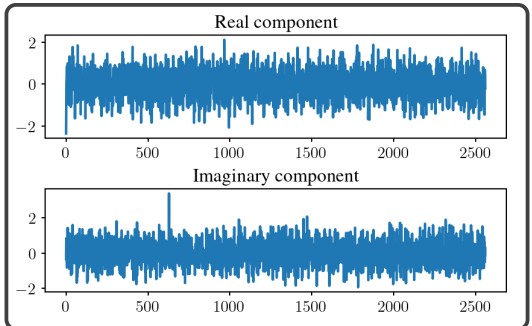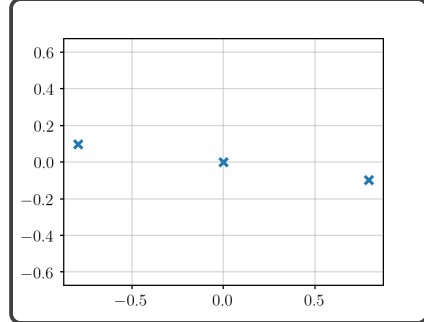

Figure 9: **Left:** A complex-valued OFDM (BPSK) source augmented with a time shift of 8 samples and with a random phase rotation plotted in the time domain. In the time domain, the discrete structure is not discernible and the time-domain waveforms visually look like Gaussian noise. **Right:** Extracting the underlying (rotated) symbols from the waveform on the left by demodulating the OFDM signal using oracle knowledge about the FFT size, cyclic prefix, and time shift.

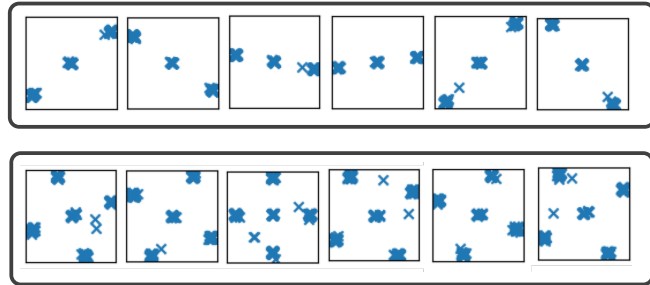

Figure 10: **Top:** Probing six generated samples from the OFDM (BPSK) diffusion model recovers the underlying BPSK constellation after time synchronization. **Bottom:** Probing six generated samples from the OFDM (QPSK) diffusion model recovers the underlying BPSK constellation after time synchronization.

The top row of Figure 10 shows the recovered constellation for six generated OFDM (BPSK) symbols and the bottom row shows the same for six generated OFDM (QPSK) signals. Note that since there are some null symbols, i.e., unused subcarriers in the frequency domain, there is an additional "constellation point" at the origin. In general the recovered constellations are clean, except for a few symbols that lie slightly off the constellation. When converting back to bits the symbols are mapped to the nearest constellation point.

### D.3 CommSignal2

The CommSignal2 dataset was recorded over-the-air and hence, unlike the synthetic datasets, we cannot probe it without knowledge of the true system parameters and signal generation model, which is not provided in [44]. These complications serve as one of our motivations to develop data-driven source separation models. In the real-world, CommSignal2 could interfere with an SOI such as a QPSK signal for which the signal generation model is known. A learned prior for CommSignal2 could help mitigate such interference.

As shown in Figure 11, we observe that the generated CommSignal2 waveforms generally display similar global temporal structure to the ground truth waveforms. To further demonstrate that our diffusion model has truly learned the statistical structure of the signal, we carry out source separation experiments and demonstrate that leveraging this diffusion model outperforms conventional and learning-based source separators.

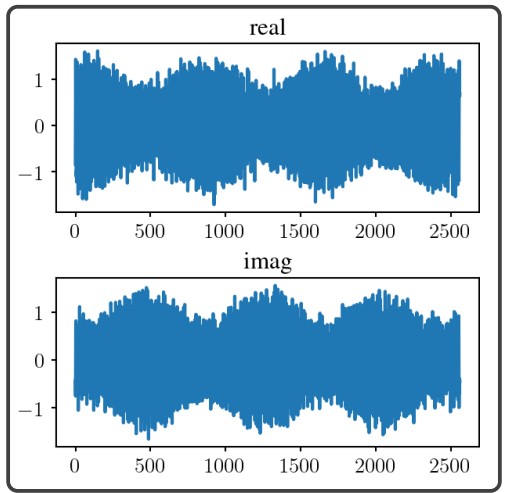 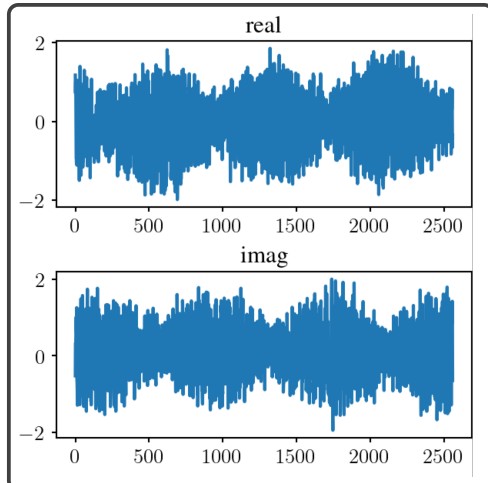

Figure 11: **Left:** Ground truth CommSignal2 waveform from the dataset. **Right:** A generated CommSignal2 waveform from the learned diffusion model.

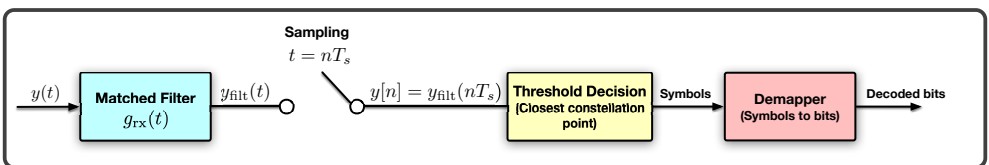

Figure 12: Block diagram of the **matched filtering demodulation** pipeline used in our work. (First symbol at $\tau_s = 8$, symbol period $T_s = 16$, as detailed in Appendix C

## E    Classical RF Interference Mitigation Techniques

### E.1    Matched Filtering

Matched filtering (MF) is a signal processing technique commonly used in the demodulation of RF waveforms. It exploits knowledge about the signal waveform to enhance the detection and recovery of the transmitted symbols/bits, and is optimal in the maximum-SNR sense for signals contaminated with additive Gaussian noise.

The basic principle involves filtering (or similarly viewed as correlating) the received sampled RF waveform with a known reference waveform called the "matched filter". The goal is to maximize the signal-to-interference-plus-noise ratio (SINR) at the filtered output, which consequently minimizes the error probability in the subsequent symbol detection when the noise is Gaussian. In the following segment, we develop the theory of the matched filter, with particular focus to our problem formulation.

Consider the baseband RRC-QPSK signal as described in Appendix C. Suppose, for the purposes of this exposition, that we adopt a simple additive white Gaussian noise (AWGN) channel model, thereby representing our received signal as

$$y(t) = \sum_p c_p \, g_{\text{tx}}(t - pT_s) + w(t) \tag{61}$$

$$= g_{\text{tx}}(t) * \sum_p c_p \, \delta(t - pT_s) + w(t), \tag{62}$$

where $c_p$ are the symbols from a QPSK constellation (see Figure 6b), $*$ denotes the convolution operator, $\delta(\cdot)$ is the dirac delta fucnction, and $w(t) \sim \mathcal{N}(0, \sigma_{\text{AWGN}}^2)$ is the additive noise in the observed signal, statistically independent of all $\{c_p\}$. Of particular interest in this formulation is the transmit pulse shaping function $g_{\text{tx}}(t)$, where we chose to use the RRC function. (This is due its favorable properties in practice; however, the optimal transmit filter choices and waveform designs

are outside the scope of this work, and we defer readers to related resources for more information on this topic [18, 36].)

At the receiver, we seek a receiver filter, $g_{\mathrm{rx}}(t)$, such that the filtered and sampled output

$$y_{\mathrm{filt}}(t) = \underbrace{g_{\mathrm{rx}}(t) * g_{\mathrm{tx}}(t)}_{:=g(t)} * \sum_p c_p \delta(t - pT_s) + g_{\mathrm{rx}}(t) * w(t) \tag{63}$$

$$y[n] = y_{\mathrm{filt}}(nT_s) = \sum_p c_p\, g((n-p)T_s) + \underbrace{\int w(\tau)\, g_{\mathrm{rx}}(nT_s - \tau)\mathrm{d}\tau}_{:=v[n]} \tag{64}$$

$$= \underbrace{c_n\, g(0)}_{:=y_s[n]} + \underbrace{\sum_{p \neq n} c_n\, g((n-p)T_s) + v[n]}_{:=y_v[n]} \tag{65}$$

would maximize the output SINR. In other words, we are looking to maximize

$$\mathrm{SINR} = \frac{\mathbb{E}\left[|y_s[n]|^2\right]}{\mathbb{E}\left[|y_v[n]|^2\right]} = \frac{\mathbb{E}\left[|c_n|^2\right]|g(0)|^2}{\mathbb{E}\left[|c_n|^2\right]\sum_{p \neq n} |g(pT_s)|^2 + \sigma_{\mathrm{AWGN}}^2 \int |G_{\mathrm{rx}}(f)|^2\mathrm{d}f} \tag{66}$$

(where $G_{\mathrm{rx}}(f)$ is the Fourier transform of $g_{\mathrm{rx}}(t)$) via an appropriate choice of $g(t)$—and thereby, $g_{\mathrm{rx}}(t)$. This can be done by finding an upper bound on the SINR that reaches equality for the appropriate filter choices. Ultimately, one such choice is $g_{\mathrm{rx}}(t) = g_{\mathrm{tx}}^*(-t)$—termed as the *matched filter*—that leads to a maximized SINR. In the case of an RRC pulse shaping function (which is real and symmetric), the matched filter is also the same RRC function.

In this work, we refer to matched filtering more broadly to also include the detector/decoding step, as shown in Figure 12. As part of the demodulation pipeline, the filtered output is sampled (as in (65)), and then mapped to the closest symbol in a predefined constellation (in the Euclidean distance sense). Finally, we can map these complex-valued symbols back to their corresponding bits to recover the underlying information. We use this as a standard demodulation/detection pipeline for our RRC-QPSK SOI following the corresponding signal separation/interference mitigation step (where applicable).

Demodulation with matched filtering is optimal for waveforms in the presence of additive Gaussian noise. However, in our signal separation problem, we consider the presence of an additive interference, which is not necessarily Gaussian. Thus, exploiting the non-Gaussian characteristics of the interference would likely lead to enhanced decoding performance.

### E.2 LMMSE Estimation

Recall that our observation model is

$$\mathbf{y} = \mathbf{s} + \kappa\mathbf{b},$$

where we assume $\mathbf{s}$ and $\mathbf{b}$ are zero-mean and that they are statistically independent. The linear minimum mean square error (LMMSE) estimator is the estimator $\widehat{\mathbf{s}} = \boldsymbol{W}_{\mathrm{LMMSE}}\mathbf{y}$, such that

$$\boldsymbol{W}_{\mathrm{LMMSE}} = \underset{\boldsymbol{W} \in \mathbb{C}^{T \times T}}{\arg\min}\, \mathbb{E}\left[\|\mathbf{s} - \boldsymbol{W}\mathbf{y}\|_2^2\right]. \tag{67}$$

In this case, the optimal linear transformation (in the sense of (67)) can be written as

$$\boldsymbol{W}_{\mathrm{LMMSE}} = \boldsymbol{C}_{sy}\, \boldsymbol{C}_{yy}^{-1} = \boldsymbol{C}_{ss}\, (\boldsymbol{C}_{ss} + \kappa^2 \boldsymbol{C}_{bb})^{-1}$$

where $\boldsymbol{C}_{sy} \coloneqq \mathbb{E}[\mathbf{s}\mathbf{y}^{\mathrm{H}}]$ corresponds to the cross-covariance between $\mathbf{s}$ and $\mathbf{y}$, $\boldsymbol{C}_{yy}, \boldsymbol{C}_{ss}, \boldsymbol{C}_{bb}$ are the auto-covariance of $\mathbf{y}, \mathbf{s}$ and $\mathbf{b}$ respectively. The second equality is obtained by statistical independence, thereby $\boldsymbol{C}_{sy} = \boldsymbol{C}_{ss}, \boldsymbol{C}_{yy} = \boldsymbol{C}_{ss} + \kappa^2 \boldsymbol{C}_{bb}$.

In relevant literature, this may also be referred to as linear MMSE receivers [50, Sec 8.3.3]. Note that we aim to mitigate the effects of an additive interference channel. For the purposes of this work, we use LMMSE as one of the baselines as a signal separation (interference mitigation) method. Thereafter, we assess performance based on the squared error between $\widehat{\mathbf{s}}$ with the ground truth $\mathbf{s}$. To obtain the underlying bits, we perform a standard matched filtering operation on the estimator $\widehat{\mathbf{s}}$.

To implement the LMMSE estimator described above, one simply requires the covariances $C_{yy}$, $C_{ss}$, $C_{bb}$. Under the problem formulation, one could potentially obtain the sample covariance, whose precision scales with the number of samples used. In the case of the synthetic signals, we assume oracle knowledge of the covariance when computing this baseline, so as to eliminate confounding effects from poor estimation of these statistics. However, for CommSignal2, we assume that the signal is zero-mean (validated with the empirical mean), and compute the autocorrelation matrix for time windows of length 2560, across 10,000 sample realizations drawn randomly from the frames provided in the dataset. The empirical covariance estimate that we used will be provided in the accompanying code/data repository.

Note that in this problem, the interference also has random time offsets, which has to be accounted for when computing or estimating the covariance of our interference signals. The works in [43, 12] address the effects of such random time offsets in detail, and discuss the suboptimality of the above LMMSE in such a setting. We acknowledge that without an additional time synchronization step, the LMMSE estimator adopted herein may be modeling the interference as time series from a stationary process.

We remark that the LMMSE estimator is optimal if the components were Gaussian. However, as digital communication signals contain some underlying discreteness and undergo unknown time-shifts, these signals are typically non-Gaussian (and often, even far from Gaussian). Hence, better performance can generally be obtained through nonlinear methods.

## F    BASIS for RF Source Separation

We use the BASIS method [5] as a learning-based benchmark for SCSS in our experiments. Similar to our method, BASIS leverages pre-trained generative models as statistical priors and does not rely on end-to-end (supervised) training with paired data. The manner in which these priors are used for separation, however differs:

1. **Soft constraint vs. hard constraint:** In the context of our problem formulation, we are interested in decomposing a mixture $\mathbf{y} = \mathbf{s} + \kappa\mathbf{b}$ into its constituent components. As described in §3.2 our proposed method recovers the components by extending a MAP estimation rule with a hard constraint given by,

$$\widehat{\mathbf{s}} = \underset{\mathbf{s}\in\mathcal{S} \text{ s.t } \mathbf{y}=\mathbf{s}+\kappa\mathbf{b}}{\arg\max}\ p_{\mathsf{s}|\mathsf{y}}(\mathbf{s}|\mathbf{y}), \tag{68}$$

where the resulting estimate of $\mathbf{b}$ is $\widehat{\mathbf{b}} = (\mathbf{y} - \widehat{\mathbf{s}})/\kappa$. The result of imposing this constraint is a MAP objective that leverages two priors but only requires optimization with a single

---

**Algorithm 2** BASIS Separation (original, as proposed in [5])

---

1: **function** SEPARATION($\mathbf{y}, \kappa, N, \{\sigma_t^2\}_{t=1}^T, \boldsymbol{\psi}_s^{(0)}, \boldsymbol{\psi}_b^{(0)}$) ▷ Specially tuned noise schedule $\{\sigma_t\}_{t=1}^T$
2:    **for** $t \leftarrow 1, T$ **do**
3:        $\boldsymbol{\psi}_s^{(t)} \leftarrow \boldsymbol{\psi}_s^{(t-1)}$, $\boldsymbol{\psi}_b^{(t)} \leftarrow \boldsymbol{\psi}_b^{(t-1)}$, $\eta_t \leftarrow f(\sigma_t^2)$                    ▷ Learning rate $\eta_t$
4:        **for** $i \leftarrow 0, N-1$ **do**
5:            $\boldsymbol{\epsilon}_s, \boldsymbol{\epsilon}_b \sim \mathcal{N}(0, \mathbf{I})$
6:            $\widehat{\mathbf{z}}_s, \widehat{\mathbf{z}}_b = r_{\phi_{\mathbf{s}}}\left(\boldsymbol{\psi}_s^{(t)}, t\right), r_{\phi_{\mathbf{b}}}\left(\boldsymbol{\psi}_t^{(t)}, t\right)$            ▷ Compute scores at noise level $\sigma_t$
7:            $\boldsymbol{\psi}_s^{(t)} \leftarrow \boldsymbol{\psi}_s^{(t)} - \eta_t \underbrace{(\widehat{\mathbf{z}}_s/\sigma_t)}_{\text{approx. score}} -\eta_t\left(\mathbf{y} - \boldsymbol{\psi}_s^{(t)} - \kappa\boldsymbol{\psi}_b^{(t)}\right)/\sigma_t^2 + \sqrt{2\eta_t}\boldsymbol{\epsilon}_s$
8:            $\boldsymbol{\psi}_b^{(t)} \leftarrow \boldsymbol{\psi}_b^{(t)} - \eta_t\left(\widehat{\mathbf{z}}_b/\sigma_t\right) - \eta_t\kappa\left(\mathbf{y} - \boldsymbol{\psi}_s^{(t)} - \kappa\boldsymbol{\psi}_b^{(t)}\right)/\sigma_t^2 + \sqrt{2\eta_t}\boldsymbol{\epsilon}_b$
9:        **end for**
10:    **end for**
11:    **return** $\widehat{\mathbf{s}} = \boldsymbol{\psi}_s^{(T)}$, $\widehat{\mathbf{b}} = \boldsymbol{\psi}_b^{(T)}$
12: **end function**

---

variable,

$$\arg\max_{\mathbf{s}\in\mathcal{S} \text{ s.t } \mathbf{y}=\mathbf{s}+\kappa\mathbf{b}} p_{\mathsf{s|y}}(\mathbf{s}|\mathbf{y}) \quad = \arg\max_{\mathbf{s}\in\mathcal{S} \text{ s.t } \mathbf{y}=\mathbf{s}+\kappa\mathbf{b}} p_{\mathsf{y|s}}(\mathbf{y}|\mathbf{s})P_{\mathsf{s}}(\mathbf{s}) \tag{69a}$$

$$= \arg\min_{\mathbf{s}\in\mathcal{S}} -\log P_{\mathsf{s}}(\mathbf{s}) - \log p_{\mathsf{b}}\left((\mathbf{y}-\mathbf{s})/\kappa\right). \tag{69b}$$

In contrast, BASIS strives to estimate the underlying components by *sampling from the posterior*. In particular, they do not enforce a hard constraint during sampling and instead assume a likelihood of the form

$$p_{\mathsf{y|s,b}}^{\text{BASIS}}(\mathbf{y}|\mathbf{s},\mathbf{b}) = \mathcal{N}\left(\mathbf{y};\mathbf{s}+\kappa\mathbf{b},\gamma^2\mathbf{I}\right), \tag{70}$$

which actually corresponds to an alternate mixture model with auxilliary noise $\mathbf{w}\sim\mathcal{N}(0,\gamma^2\mathbf{I})$ such that,

$$\mathbf{y} = \mathbf{s} + \kappa\mathbf{b} + \mathbf{w}. \tag{71}$$

Thus, the estimates $\widehat{\mathbf{s}}^{\text{BASIS}}$ and $\widehat{\mathbf{b}}^{\text{BASIS}}$ are obtained by sampling from the posterior,

$$p_{\mathsf{s,b|y}}^{\text{BASIS}}(\mathbf{s},\mathbf{b}|\mathbf{y}) = p_{\mathsf{y|s,b}}^{\text{BASIS}}(\mathbf{y}|\mathbf{s},\mathbf{b})\,P_{\mathsf{s}}(\mathbf{s})p_{\mathsf{b}}(\mathbf{b}). \tag{72}$$

Notice that sampling from this posterior requires computing *two separate estimates* due to the soft constraint in (71).

2. **Annealed Langevin dynamics vs. randomized levels of Gaussian smoothing:** The aforementioned BASIS estimates satisfy $\mathbf{y} = \widehat{\mathbf{s}}^{\text{BASIS}} + \kappa\widehat{\mathbf{b}}^{\text{BASIS}}$ only when $\gamma\to 0$ in the soft constraint in (71) enforced via the likelihood term. As shown in Algorithm 2, the idea behind BASIS is to leverage annealed Langevin dynamics sampling to sample from the posterior by slowly decreasing the value of $\gamma$ towards zero by *tuning a specially chosen noise schedule*. While generally a separate noise schedule is required per prior, the authors suggest sharing the same schedule across both priors. Apart from tuning the noise schedule, the learning rate schedule is another hyperparameter that must also be tuned. In contrast, our algorithm leverages the existing noise schedule from pre-trained diffusion models in a randomized fashion *without any additional tuning*.

3. **Optimization via sampling vs. optimization via gradient descent:** Starting with initializations $\boldsymbol{\psi}_s^{(0)}$ and $\boldsymbol{\psi}_b^{(0)}$ for the SOI and interference respectively, BASIS refines the estimate such that $\boldsymbol{\psi}_s^{(t)}$ approximates a sample drawn from the distribution of the smoothened source $p_{\tilde{\mathsf{s}}_t}$. Langevin dynamics leverages the approximate score of $p_{\tilde{\mathsf{s}}_t}$, to update the estimate by *sampling from higher density regions*. A similar update is performed for the interference. The estimates are constrained to satisfy (71) through the Gaussian likelihood.

In contrast, as described in Algorithm 1 (see §3.2.2), we obtain an estimate $\boldsymbol{\theta}^{(t)}$ of the unsmoothed SOI for each randomly drawn noise level. The hard constraint allows us to readily get an estimate for the interference as $(y - \boldsymbol{\theta}^{(t)})/\kappa$. As described in §3.2.2, we then update the SOI estimate using gradient descent, with a rule guided by (69b), using the *gradients of the log densities* with respect to $\boldsymbol{\theta}$ (which is functionally equivalent to the scaled scores). The ideal update gradient is,

$$\nabla_{\boldsymbol{\theta}}\mathcal{L}(\boldsymbol{\theta}) := -\mathbb{E}_{\mathsf{t},\mathsf{z}_t}\left[\sqrt{\alpha_t}\,S_{\tilde{\mathsf{s}}_t}\left(\tilde{\mathbf{s}}_t\left(\boldsymbol{\theta}\right)\right)\right] + \frac{\omega}{\kappa}\mathbb{E}_{\mathsf{u},\mathsf{z}_u}\left[\sqrt{\alpha_u}S_{\tilde{\mathsf{b}}_u}\left(\tilde{\mathbf{b}}_u\left(\boldsymbol{\theta},\mathbf{y}\right)\right)\right]. \tag{73}$$

We first implement BASIS in its original form, with our best attempt at finding an appropriate learning rate. As shown in Figure 13, on QPSK (SOI) + OFDM (QPSK) mixture, this leads to worse performance than matched filtering—likely due to poor matching of the observation model in (71). Note that we are limited in the annealing schedule by the set of possible noise levels, based on the training setup of our diffusion models. Thus, we use the diffusion model training noise schedule discretized into $T = 50$ timesteps (see §5.2) and additionally use $N = 100$. We seek to use the largest set of noise levels possible, consistent with approaches in most Langevin sampling based works. We tuned the learning rate schedule to the best of our abilities and found that $\eta_t = f(\sigma_t^2) = (2\text{e}{-}8)\sigma_t^2/\sigma_1^2$, gave the best results. All curves use an analytical score model for the SOI.

To make an even more conservative comparison (w.r.t. our proposed method), we replace the posterior using (72), which corresponds to the formulation in (71), with a hard constraint as in our setup. This

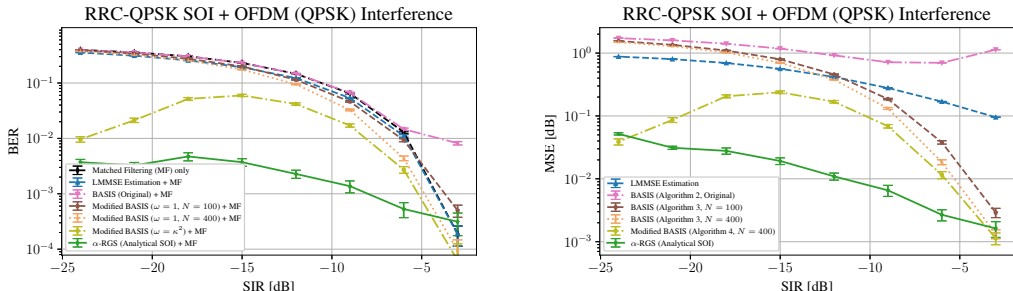

Figure 13: Comparing BASIS and Modified BASIS against conventional RF source separation methods. The vanilla BASIS algorithm performs worse than MF, while the Modified BASIS algorithm, which is inspired by our usage of an $\alpha$-posterior, is able to beat LMMSE. Still, our method outperforms all baselines.

---

**Algorithm 3** BASIS Separation, using (69b)

---

1: **function** SEPARATION($\mathbf{y}, \kappa, N, \{\sigma_t^2\}_{t=1}^T, \boldsymbol{\theta}^{(0)}$)     ▷ Specially tuned noise schedule $\{\sigma_t\}_{t=1}^T$
2:     **for** $t \leftarrow 1, T$ **do**
3:         $\boldsymbol{\theta}^{(t)} \leftarrow \boldsymbol{\theta}^{(t-1)}, \eta_t \leftarrow f(\sigma_t^2)$     ▷ Learning rate $\eta_t$
4:         **for** $i \leftarrow 0, N-1$ **do**
5:             $\boldsymbol{\epsilon} \sim \mathcal{N}(0, \mathbf{I})$
6:             $\widehat{\mathbf{z}}_s, \widehat{\mathbf{z}}_b = r_{\phi_\mathbf{s}}\left(\boldsymbol{\theta}^{(t)}, t\right), r_{\phi_\mathbf{b}}\left(\left(\mathbf{y} - \boldsymbol{\theta}^{(t)}\right)/\kappa, t\right)$   ▷ Compute scores at noise level $\sigma_t$
7:             $\boldsymbol{\theta}^{(t)} \leftarrow \boldsymbol{\theta}^{(t)} \underbrace{-\eta_t\left(\widehat{\mathbf{z}}_s/\sigma_t\right) + \frac{1}{\kappa}\eta_t\left(\widehat{\mathbf{z}}_b/\sigma_t\right)}_{\text{from MAP objective}} + \sqrt{2\eta_t}\boldsymbol{\epsilon}$
8:         **end for**
9:     **end for**
10:     **return** $\widehat{\mathbf{s}} = \boldsymbol{\theta}^{(T)}, \widehat{\mathbf{b}} = \left(\mathbf{y} - \boldsymbol{\theta}^{(T)}\right)/\kappa$
11: **end function**

---

**Algorithm 4** Modified BASIS Separation by introducing $\alpha$-posterior

---

1: **function** SEPARATION($\mathbf{y}, \kappa, N, \{\sigma_t^2\}_{t=1}^T, \boldsymbol{\theta}^{(0)}$)     ▷ Specially tuned noise schedule $\{\sigma_t\}_{t=1}^T$
2:     **for** $t \leftarrow 1, T$ **do**
3:         $\boldsymbol{\theta}^{(t)} \leftarrow \boldsymbol{\theta}^{(t-1)}, \eta_t \leftarrow f(\sigma_t^2)$     ▷ Learning rate $\eta_t$
4:         **for** $i \leftarrow 0, N-1$ **do**
5:             $\boldsymbol{\epsilon} \sim \mathcal{N}(0, \mathbf{I})$
6:             $\widehat{\mathbf{z}}_s, \widehat{\mathbf{z}}_b = r_{\phi_\mathbf{s}}\left(\boldsymbol{\theta}^{(t)}, t\right), r_{\phi_\mathbf{b}}\left(\left(\mathbf{y} - \boldsymbol{\theta}^{(t)}\right)/\kappa, t\right)$   ▷ Compute scores at noise level $\sigma_t$
7:             $\boldsymbol{\theta}^{(t)} \leftarrow \boldsymbol{\theta}^{(t)} \underbrace{-\eta_t\left(\widehat{\mathbf{z}}_s/\sigma_t\right) + \frac{\omega}{\kappa}\eta_t\left(\widehat{\mathbf{z}}_b/\sigma_t\right)}_{\text{from MAP objective with }\alpha\text{-posterior}} + \sqrt{2\eta_t}\boldsymbol{\epsilon}$
8:         **end for**
9:     **end for**
10:     **return** $\widehat{\mathbf{s}} = \boldsymbol{\theta}^{(T)}, \widehat{\mathbf{b}} = \left(\mathbf{y} - \boldsymbol{\theta}^{(T)}\right)/\kappa$
11: **end function**

---

method, as further described in Algorithm 3, leads to better performance, even outperforming LMMSE in some cases. We also experiment with $N = 400$ (for a total of $20,000$ iterations), and do not observe significant gains. Empirically we found this method to be typically stuck in local optimum which is in line with our characterization earlier in §4.

Finally, we augment the BASIS algorithm with an $\alpha$-posterior with $\alpha = \omega$, as described in Algorithm 4. As shown in Figure 13, the $\alpha$-posterior augmented method obtains better performance than vanilla BASIS, but is generally unable to reach the performance of our method. We emphasize again that in contrast to our method, the aforementioned BASIS-based methods follow a prescribed annealing

schedule, that can require cumbersome tuning for optimal performance. The implementations of Algorithm 3 and Algorithm 4 both use a learning rate schedule of $\eta_t = f(\sigma_t^2) = (2\mathrm{e}{-}6)\sigma_t^2/\sigma_1^2$

# G   Complete Results

In this section we will provide the complete results of our study. We first present another baseline based on simulating the reverse diffusion process [21] as a denoiser, that complements the baselines already described in §5.2.

**Reverse Diffusion.**   Given a mixture $\mathbf{y}$, we interpret the SOI as (scaled) additive noise on top of the interference $\mathbf{b}$. Since the reverse diffusion process can be interpreted as an iterative denoiser (see §2.2), we run a small chain of reverse diffusion for 10 timesteps starting at a noise variance of 0.005 and ending with variance of 0.0001, which was chosen based on different trials. Similar ideas have been used in prior works involving inverse problems for images [31, 51]. We note that it is generally cumbersome to find the optimal reverse diffusion noise range and it might in fact be dependent on the specific mixture's SIR.

**Extracting the underlying bits.**   Our primary metric for assessing the quality of the estimated SOI is through the BER between the estimated bits and the truly encoded bits. The estimate of the SOI typically equals the true SOI with a very small amount of noise due to the continuous nature of the optimization problem. Thus, we can expect the underlying constellation to contain some symbols concentrated around, but not exactly at the constellation points. This can be modeled as the true SOI corrupted with a small amount of Gaussian noise, for which MF is the optimal technique to decode the bits by mapping the symbols to the closest constellation point (see Section E.1).

**The analytical SOI score.**   We use an RRC-QPSK signal as the SOI across all our source separation experiments. As detailed in Section B.2, the score for a smoothened QPSK source can be analytically computed via Proposition 3, where it is more amenable to model and compute it in the symbol space (i.e., as i.i.d. symbols). Nevertheless, for the problem of separating the SOI from an interference source, we have to consider the components jointly in the time domain. Thus, we relate the time-domain representation to the symbols via $\mathbf{s} = \boldsymbol{H}\mathbf{a}$, where $\boldsymbol{H}$ represents the RRC filter matrix and $\mathbf{a}$ is a vector of symbols. To compute this analytical score, we smooth the symbols via a Gaussian smoothing model at noise level corresponding to timestep $t$,

$$\tilde{\mathbf{s}}_t = \boldsymbol{H}\tilde{\mathbf{a}}_t := \boldsymbol{H}(\gamma_t \mathbf{a} + \sigma_t \mathbf{z}_s). \tag{74}$$

With this relation, we express the score of the smoothened source as,

$$\nabla_{\tilde{\mathbf{s}}_t} \log p_{\tilde{\mathbf{s}}_t}(\tilde{\mathbf{s}}_t) = \boldsymbol{H} \cdot \nabla_{\tilde{\mathbf{a}}_t} \log p_{\tilde{\mathbf{a}}_t}(\tilde{\mathbf{a}}_t) \tag{75}$$

$$= \boldsymbol{H} \cdot \left( \frac{1}{\sigma_t^2} \left( -\tilde{\mathbf{a}}_t + \sum_{\mathbf{a} \in \mathcal{A}^K} \mathbf{a} \odot \phi_t\left(\mathbf{a}; \tilde{\mathbf{a}}_t\right) \right) \right) \tag{76}$$

$$\approx \frac{1}{\sigma_t^2} \left( -\tilde{\mathbf{s}}_t + \boldsymbol{H} \sum_{\mathbf{a} \in \mathcal{A}^K} \mathbf{a} \odot \phi_t\left(\mathbf{a}; \boldsymbol{H}^\dagger \tilde{\mathbf{s}}_t\right) \right). \tag{77}$$

where $\odot$ represents element-wise product, $\phi_t$ is the softmax-like operator defined in (34) that is applied element-wise here, and $\boldsymbol{H}^\dagger$ is the pseudo-inverse of $\boldsymbol{H}$. Note that (76) is obtained by applying (33) to a vector of i.i.d. smoothened QPSK symbols. In our implementation, the estimate of these smoothened symbols $\tilde{\mathbf{a}}_t$ is obtained by reversing the RRC filter using $\boldsymbol{H}^\dagger$, as in (77).

On the other hand, our RRC-QPSK diffusion model is trained directly on the time-domain waveform, and can be used directly to separate the SOI waveform from the mixture without conversion to the symbol domain. This once again sheds light on the practicality of using data-driven methods to circumvent otherwise computationally challenging statistical modeling technical problems.

**Results.**   Figures 14 and 15 show the complete source separation results for mixtures with OFDM and CommSignal2 as interference, respectively. Our model that uses an analytical SOI score for the SOI and a diffusion-based score for the interference generally performs the best and outperforms all baselines. Furthermore, we show that using a learned SOI score still outperforms all baselines in

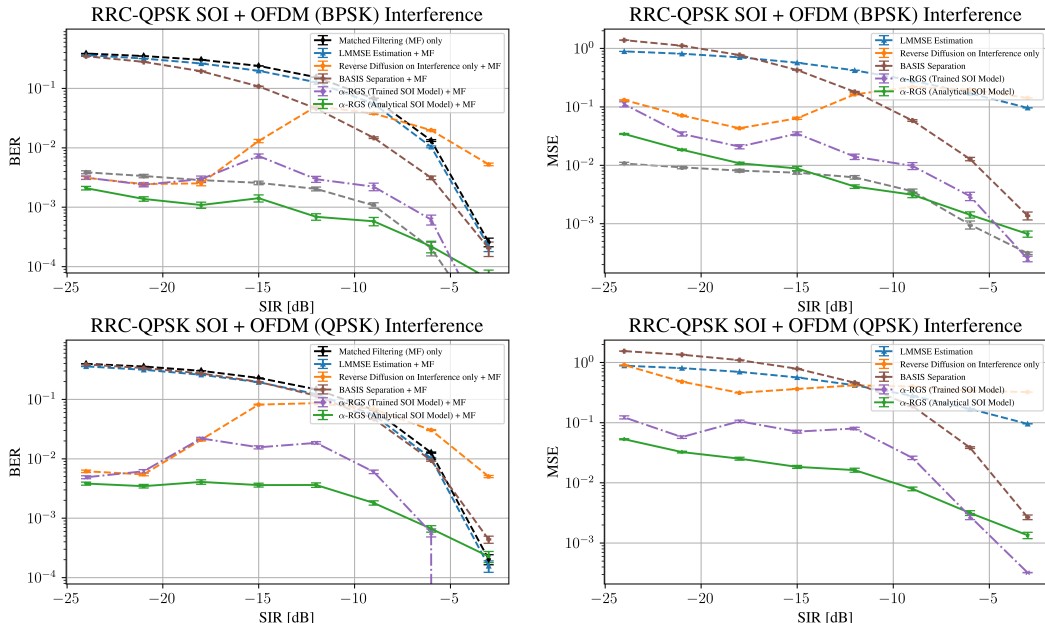

Figure 14: Comparing our method against various baselines on separating (**top**) RRC-QPSK + OFDM (BPSK) mixtures and (**bottom**) RRC-QPSK + OFDM (QPSK) mixtures. Our model that uses an analytical SOI score outperforms all baselines in terms of BER. Reverse diffusion is competitive at low SIRs since the interference dominates the mixture in this regime and hence iterative denoising to separate the SOI is effective. The trained SOI diffusion models significantly outperform all baselines at high SIR. The reason the analytical SOI performs slightly worse is due to the approximations we make in (77).

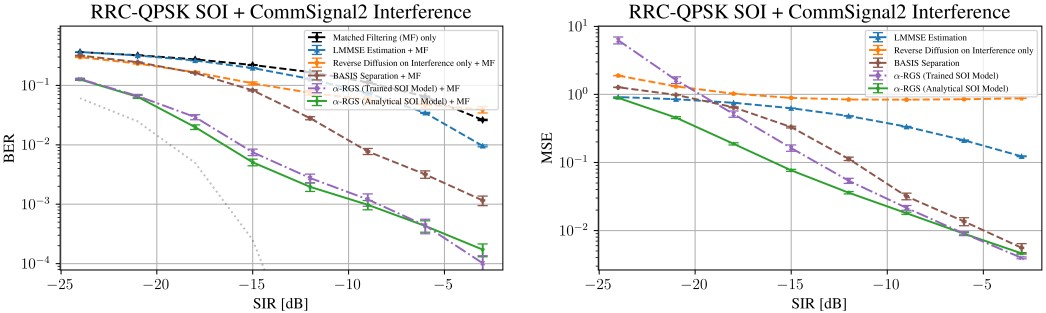

Figure 15: Comparing our method against various baselines on separating RRC-QPSK + CommSignal2 mixtures. Our method significantly outperforms all baselines in terms of BER. The large MSE at low SIRs suggests that there is background noise present in the CommSignal2 source, which is amplified at lower SIR (large $\kappa$). We try to estimate the amount of noise and model it as additive Gaussian noise. Accounting for this noise could presumably lead to a lower bound on the BER, shown by dotted black line on the left.

terms of BER, despite the slight degradation. The trained SOI score models consistently outperform all methods at high SIR in terms of BER since the SOI diffusion model was trained with RRC-QPSK samples as opposed to the approximations made in (77) in implementing the analytical score.

Our method also outperforms baselines in terms of MSE for OFDM mixtures as shown in Figure 14. The performance at low SIR in the context of CommSignal2 mixtures is not the same. As shown in Figure 15, the large MSE at low SIRs suggest that there is background noise present in the CommSignal2 sources that is amplified for large values of $\kappa$, i.e., the interference is actually of the form $\mathbf{b} + \mathbf{w}$ for some background noise $\mathbf{w}$. We validated that this was indeed the case, by visualizing

samples in both the time-domain and frequency domain using the RF challenge demo notebook[10]. As shown in Figure 16, we noticed segments of lower magnitude at the start and end, which we believe to be background noise that is not part of intended communication signal. During source separation, this presumably results in a noisier estimate of the SOI in comparison to mixtures with no additional background noise. We estimated the SNR to be 16.9 dB by averaging across multiple samples. The dotted black curve on the left of Figure 15 is a presumable lower bound on the BER by accounting for the magnitude of the background noise and modeling it as additive white Gaussian noise.

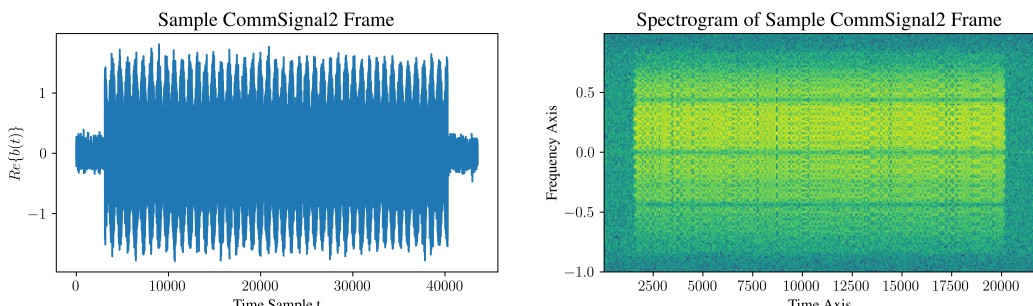

Figure 16: Time-domain and frequency-domain plots of a single frame from the CommSignal2 dataset. **Left:** In the time-domain we observe segments of lower magnitude at the start and end, which we believe to be background noise. **Right:** In the frequency-domain we observe the bandlimited communication signal with spectrally flat features in the regions we identified as background noise.

Nevertheless, across all experiments, we demonstrate that our method sets a new state-of-the-art for applications such as interference mitigation where the interference can be learned from data recordings and the desired information can be decoded with knowledge of the SOI demodulation pipeline.

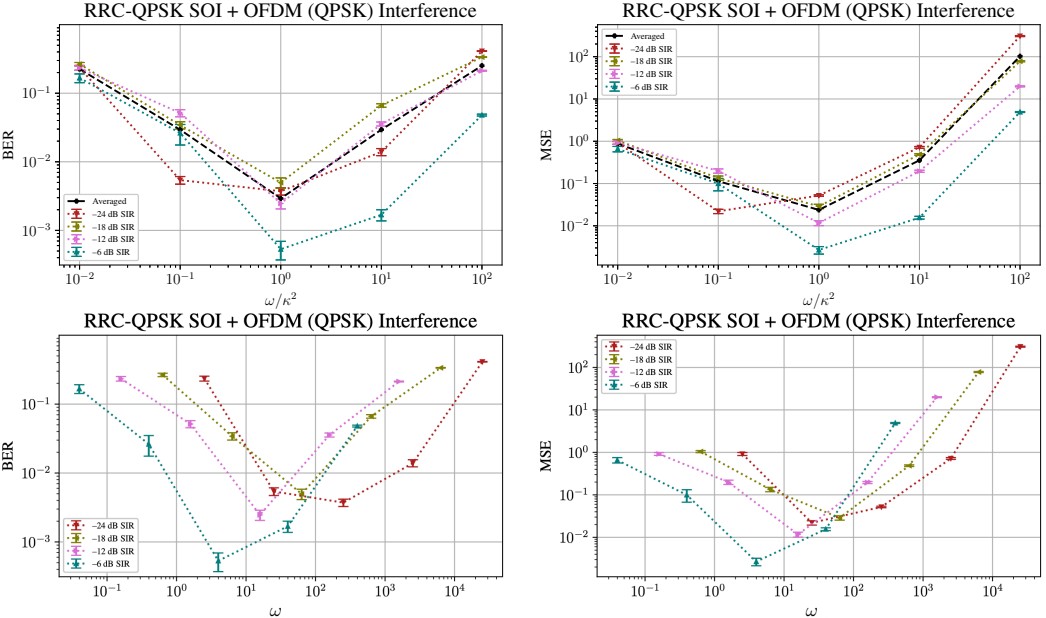

Figure 17: **Top:** BER and MSE versus $\omega/\kappa^2$ for different SIR levels. Evidently, a good choice of $\omega$, on average, across different noise levels is $\omega = \kappa^2$. **Bottom:** By looking at individual SIRs we see that the minimum BER and MSE is achieved when $\omega$ increases with increasing $\kappa^2$ (decreasing SIR), visualized using a log scale on the x-axis.

---

[10]https://github.com/RFChallenge/rfchallenge_singlechannel_starter/blob/main/notebook/Demo.ipynb

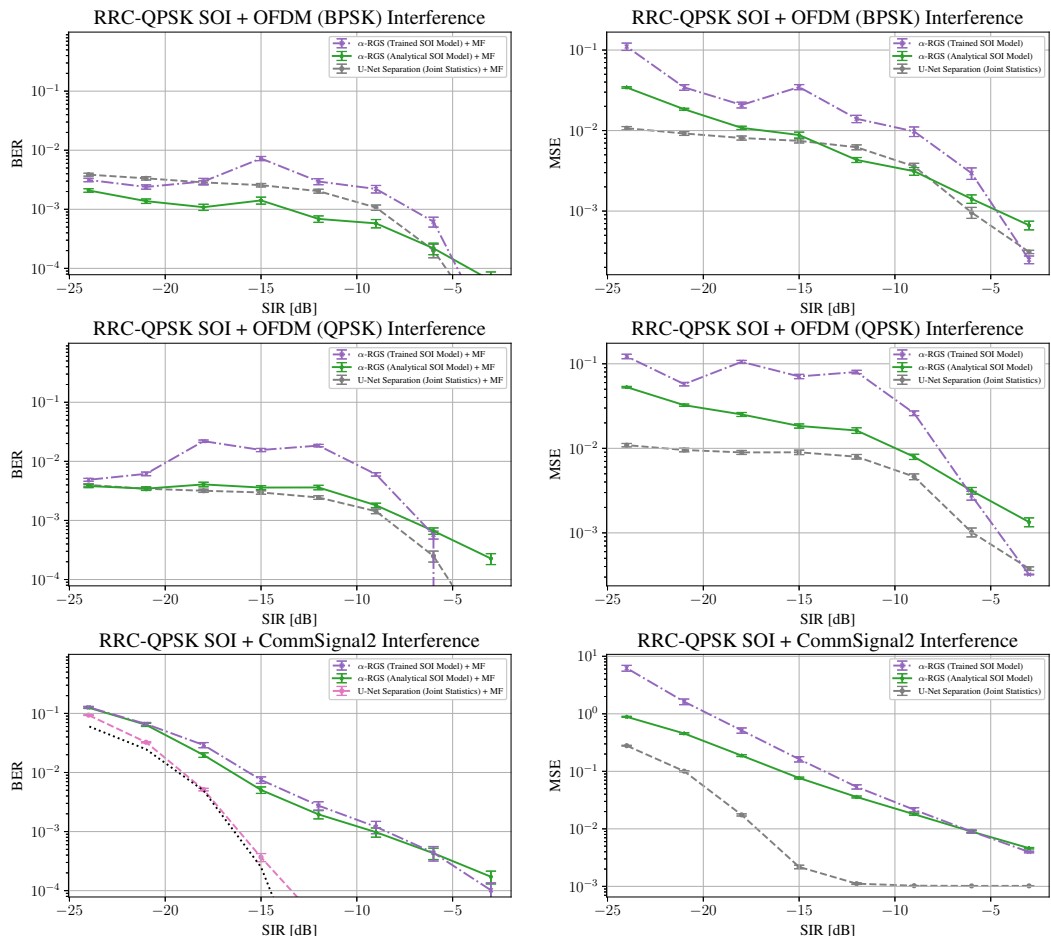

Figure 18: Comparing our method against a supervised learning setup that trains a UNet on paired data samples with an $\ell_2$ loss. We show that we are able to achieve competitive BERs, and even outperform the supervised method at certain SIRs. Our method is particularly competitive in the challenging strong interference regime (low SIR), demonstrating the fidelity of our trained interference diffusion models. The gap is larger for CommSignal2 mixtures as the supervised method is presumably able to leverage knowledge of the joint statistics between the SOI and interference to effectively deal with the background noise in the interference signal.

**Choice of $\omega$.** We numerically verify that $\omega = \kappa^2$ is a good choice of the $\alpha$-posterior term, To this end, we first find a suitable order of magnitude for $\omega$, by varying $\omega/\kappa^2$ between $10^{-2}$ and $10^2$ across different noise levels. As shown in the top row in Figure 17, on average, the minimum BER and MSE is achieved when $\omega = \kappa^2$. We additionally validate that it is beneficial to adapt $\omega$ as the SIR changes by varying $\omega$ and studying the BER and MSE curves at individual noise levels. As shown in the bottom row of Figure 17, we observe that it $\omega$ should increase with $\kappa^2$ to achieve good results.

**Comparing with supervised methods.** As motivated in §1, we are interested in leveraging independently trained priors in our source separation setup. Nevertheless, we compare against a supervised setup that learns to separate mixtures end-to-end by training a UNet on *paired* data. We train three supervised models based on the recent work in [43], using their open-sourced training code[11]. These models are trained with an $\ell_2$ loss. As such, we should expect the MSE performance to be better than our unsupervised approach, which is indeed the case across all the mixtures as shown in Figure 18. However, we notice that our method, which uses an analytical SOI score, is able to perform similarly to the UNet and even better in terms of BER for some SIRs, especially in the OFDM interference setting. Furthermore, in the challenging low SIR (strong interference regime) our method performs

---

[11]https://github.com/RFChallenge/SCSS_CSGaussian

well, showing that our interference diffusion models were able to learn the underlying statistical structures. Thus, if the mixture model changes in the future we can re-use our priors whereas the supervised approached could require an entire change to the architecture and additional training. We can potentially drive the BER in the CommSignal2 setting by modeling the background noise in these signals either through a different mixture model or through additional priors might help in driving down the BER. This will be a focus of future work.

**Computation Time.**   The inference time is as of now slightly longer than BASIS. To separate a single mixture, our method that uses $N = 20,000$ iterations takes 328 seconds on average, whereas BASIS takes 284 seconds for the same number of iterations. Meanwhile, the UNet only requires one forward pass through the model and can separate a mixture in less than one second. Our implementation was not optimized for computation time, but there may be strategies to speed up our algorithm in practice, which will be a focus of future work.

**Code.**   The code for reproducing our results can be found at `https://github.com/tkj516/score_based_source_separation` and is also linked on our project webpage `https://alpha-rgs.github.io`.

