\mathsf{s} + \sigma_t \mathsf{z}}(\gamma_t \mathsf{s} + \sigma_t \mathsf{z}_s = \tilde{s}_t(s)), \tag{B.29}$$

Our goal, to this end, is to leverage Lemma 1 to compute the score, thereby requiring an expression for the conditional expectation in (B.16). Since $\mathsf{s}$ and $\mathsf{z}_s$ are independent, (B.29) can be written as a convolution between two distributions,

$$p_{\tilde{s}_t}(\tilde{s}_t(s)) = (p_{\mathsf{s}'} * p_{\mathsf{z}_{\sigma_t}})(\mathsf{s}' + \mathsf{z}_{\sigma_t} = \tilde{s}_t(s)), \tag{B.30}$$

where

$$\mathsf{s}' := \gamma_t \mathsf{s} \quad \text{and} \quad \mathsf{z}_{\sigma_t} := \sigma_t \mathsf{z}_s. \tag{B.31}$$

In order to compute the conditional expectation, we first show that,

$$p_{\mathsf{s}'+\mathsf{z}_{\sigma_t}|\mathsf{s}'}(\mathsf{s}' + \mathsf{z}_{\sigma_t} = \tilde{s}_t(s)|\mathsf{s}' = a')p_{\mathsf{s}'}(\mathsf{s}' = a')$$
$$= \frac{1}{Z(\tilde{s}_t(s))} \left[ \lambda\, c_1(\tilde{s}_t(s)) \mathcal{N}\left( a; \frac{\gamma_t^2 \sigma_1^2 \tilde{s}_t(s) + \gamma_t \sigma_t^2 \mu_1}{\gamma_t^2 \sigma_1^2 + \sigma_t^2}, \frac{\gamma_t^2 \sigma_t^2 \sigma_1^2}{\gamma_t^2 \sigma_1^2 + \sigma_t^2} \right) \right.$$
$$\left. + (1-\lambda)\, c_2(\tilde{s}_t(s)) \mathcal{N}\left( a; \frac{\gamma_t^2 \sigma_2^2 \tilde{s}_t(s) + \gamma_t \sigma_t^2 \mu_2}{\gamma_t^2 \sigma_2^2 + \sigma_t^2}, \frac{\gamma_t^2 \sigma_t^2 \sigma_2^2}{\gamma_t^2 \sigma_2^2 + \sigma_t^2} \right) \right], \tag{B.32}$$

where

$$c_1(\tilde{s}_t(s)) := \mathcal{N}(\tilde{s}_t(s); \gamma_t \mu_1, \gamma_t^2 \sigma_1^2 + \sigma_t^2),$$
$$c_2(\tilde{s}_t(s)) := \mathcal{N}(\tilde{s}_t(s); \gamma_t \mu_2, \gamma_t^2 \sigma_2^2 + \sigma_t^2),$$
$$Z(\tilde{s}_t(s)) := \lambda\, c_1(\tilde{s}_t(s)) + (1-\lambda)\, c_2(\tilde{s}_t(s)).$$

We start by deriving the density functions for the random variables in (B.31). Since the convolution between two Gaussians is a Gaussian,

$$p_{\mathsf{s}'}(a) = \lambda \mathcal{N}(a; \gamma_t \mu_1, \gamma_t^2 \sigma_1^2) + (1-\lambda) \mathcal{N}(a; \gamma_t \mu_2, \gamma_t^2 \sigma_2^2), \tag{B.33}$$
$$p_{\mathsf{z}_{\sigma_t}}(w) = \mathcal{N}(w; 0, \sigma_t^2), \tag{B.34}$$

from which it follows, for example through the identities derived in [2, Sec 1]), that

$$p_{\tilde{s}_t}(\tilde{s}_t(s)) = \lambda \mathcal{N}(\tilde{s}_t(s); \gamma_t \mu_1, \gamma_t^2 \sigma_1^2 + \sigma_t^2) + (1-\lambda) \mathcal{N}(\tilde{s}_t(s); \gamma_t \mu_2, \gamma_t^2 \sigma_2^2 + \sigma_t^2). \tag{B.35}$$

We conclude the derivation of the conditional distribution by using Bayes' rule,

$$p_{\mathsf{s}'|\mathsf{s}'+\mathsf{z}_{\sigma_t}}(\mathsf{s}' = a|\mathsf{s}' + \mathsf{z}_{\sigma_t} = \tilde{s}_t(s)) = \frac{p_{\mathsf{s}'+\mathsf{z}_{\sigma_t}|\mathsf{s}'}(\mathsf{s}' + \mathsf{z}_{\sigma_t} = \tilde{s}_t(s)|\mathsf{s}' = a)p_{\mathsf{s}'}(\mathsf{s}' = a)}{\int p_{\mathsf{s}'+\mathsf{z}_{\sigma_t}|\