# OpenReview forum: "Score-based Source Separation with Applications to Digital Communication Signals"
_NeurIPS.cc/2023/Conference — NeurIPS 2023 poster_

### Official Review · Reviewer_AhTo · 2023-06-17

**Soundness:** 3 good
**Presentation:** 3 good
**Contribution:** 3 good
**Rating:** 7
**Confidence:** 3

**Summary:**

1.	In this paper authors proposed a method for separating superimposed sources using diffusion based generative models. The proposed model derived a new objective function based on maximum a posterior and alpha posterior.
2.	The application of the proposed work is clearly mentioned by considering the existing components in the system, The data driven score based single-channel source separation technique are proposed for signal septation which perform superior compared to the conventional methods.
3.	The contribution of the proposed work are mainly.
a.	Bayesian method for single-channel source separation is used.
b.	Score Distillation Sampling (SDS) has got good results of local extrema loss.
c.	The proposed method performs well compared to the signal processing and annealed Langevin-dynamics-based approaches for RF source separation method and the results are encouraging.
4.	The results presented in this paper are highly encouraging.


**Strengths:**

Paper is very well organized.

**Weaknesses:**

The experimental set set up should be clearly explained.

**Questions:**

1.	Justify randomizing across multiple noise levels, the local extrema of loss.
2.	What is the value of k is chosen; relative scaling coefficient between the two signals
3.	Explain the experimental set up used to derive results presented in Figure 1.
4.	Increase the resolution of the Figure 2 and explain it in detail.
5.  A detailed explanation of the diffusion model is required. papers related to diffusion model, and it can be applied to the proposed method need to be elaborated more.
6.	Why the probabilities are converted to negative in the equation 6b justify it.
7.	A Comparison is required how the computational complexity of the proposed work is reduced compared to the conventional methods.
8.	Train and test ratio used for the implementation is 90:10 is there any specific reason for it.


**Limitations:**

The author need to address the questions and revise the paper and submit it back.

---

> ### Author Rebuttal · Authors · 2023-08-08
>
> We thank the reviewer for the feedback and comments. We will address the raised questions and concerns here.
>
> **Regarding the reviewer’s comments under “weaknesses”:**
> Due to space considerations, we were not able to include all details in the main manuscript, but we have included them in the supplementary material. Appendix C provides details about the datasets and RF terminology. Appendix D provides architecture, implementation and training details about the diffusion models we used in our experiments. Appendix E provides insight into the classical baselines of matched filtering and LMMSE. Appendix F provides theoretical and implementation details about the BASIS algorithm. Appendix G presents our source separation experimental setup with a detailed overview of the results. Our intention with these multiple appendices is to help readers, and provide a paper that is as self-contained as possible,
>
> **Regarding the questions:**
> 1. **Multiple noise levels:** The reason for using multiple noise noise levels in a randomized fashion is related to the dynamics of the diffusion process. Adding noise at different levels allows the optimization algorithm to explore the landscape of the distribution at different resolutions. At low noise levels, the distribution might remain peaky and an iterative gradient-based algorithm could get stuck at suboptimal local minima (the valley between two peaks). On the other hand, at larger noise levels, the distribution naturally appears more smoothened and allows for the optimization routine to escape from local minima. Hence, using multiple noise levels allows us to trade off between exploration of the optimization landscape on one hand, and resolving the solution with a sufficiently high-resolution landscape on the other hand. Please note that this is explained in Section 3.2.1 and 3.2.2 (line 164) and Section 4 of the paper. *In the final version of the manuscript, we will better underscore these explanations.*
> 2. **Choice of $\kappa$:** In our source separation experiments we vary the relative scaling of the two components. As mentioned on line 301 and as shown in Figure 3, SIR ranges from $-24$ dB to $-3$ dB. To map this SIR range back to the range of $\kappa$s, the mapping on line 295 can be used. For example, given unit power sources $s$ and $b$, to construct a mixture with SIR -24 dB, the scaling factor required is $\kappa =15.85$.
> 3. **Experimental setup for Figure 1:** As stated in line 207, we consider two independent sources — $p_{\textsf{s}}(s)=0.5$ where $s \in \\{-1, +1\\}$ and $p_{\textsf{b}}(b)=0.25$ where $b \in \\{-2/\sqrt{20}, +2/\sqrt{20}, -6/\sqrt{20}, +6/\sqrt{20}\\}$. The joint distribution $p_{\textsf{s}, \textsf{b}}(s, b)$ has 8 equiprobable modes as shown in the leftmost plot of Figure 1. We randomly choose $s$ and $b$ to construct a mixture $y = s + \kappa b$ where $\kappa=15.85$, which corresponds to an SIR of $-24$ dB. If one plotted the line $y = s + 15.85 b$ in the $s$-$b$ plane it would appear as shown in Figure 1, with an intersection at the mode corresponding to the component values $(s, b) = (+1, -2/\sqrt{20})$. As described in Section 4, our algorithm asymptotically minimizes (14) in the limit of a large number of iterations. The central plot of Figure 1, plots (14) with $\omega=1$. Notice that the loss function is quite peaky and, especially, the incorrect mode $s=-1$ is pronounced even when the Gaussian smoothing is quite large. In the rightmost plot we demonstrate that with a suitable choice of $\omega=\kappa^2$, the optimization landscape can be made more amenable to gradient descent techniques, thus allowing convergence to the desired mode $s=+1$. *With the additional page provided in the final manuscript, we will make this explanation more clear.*
> 4. **Enlarging Figure 2**: *We will enlarge Figure 2 in the final manuscript as much as possible.* Due to space considerations, we have deferred the relevant terminology required to understand Figure 2 in greater detail to Appendix C in the supplementary.
> 5. **Literature review of diffusion models:** Thank you for this suggestion. *We will provide a more thorough review of diffusion models in the final manuscript/appendix.*
> 6. **Negative probabilities:** From (6a) to (6b) we are interested in converting a maximization problem to a minimization problem. Since the solution to a maximization problem is equivalent to minimizing the negated objective, the probabilities now have a negative sign.
> 7. **Computational complexity:** The primary objective in this study is to introduce a novel algorithm for source separation based on independently trained priors. While optimizing for the computation time is out of the scope of this paper, we make efforts to keep the implementation as efficient as possible (please take a look at the attached code) and compare the runtime of our algorithm to BASIS in Appendix G of the supplementary. *Nonetheless, we appreciate the suggestion of the reviewer and will look into optimizing the implementation for future work on the topic.  We will also move the former runtime comparisons to the results section in the final manuscript.*
> 8. **Train-test split:** The 90-10 train-test split is a standard choice for splitting the RF Challenge dataset samples (as is done in the example notebooks from the challenge’s GitHub page). We decided to use the same split in the example notebooks for our data as well. While optimizing the training and test ratio could potentially yield improved outcomes, our primary objective in this study is to introduce the novel $alpha$-RGS algorithm. As a result, our focus remains on showcasing the algorithm's potential rather than extensively fine-tuning all implementation details for optimal performance.

---

### Official Review · Reviewer_D4rU · 2023-06-28

**Soundness:** 3 good
**Presentation:** 3 good
**Contribution:** 3 good
**Rating:** 7
**Confidence:** 4

**Summary:**

This paper proposed a new Bayesian method to separate a mixture of two signals $\mathbf{s}$ and $\mathbf{b}$ from the mixture signal $\mathbf{y}=\mathbf{s}+\kappa \mathbf{b}$ where $\kappa \in \mathbb{R}_+$ is known. The new method leverages the score from pre-trained diffusion models to extend MAP estimation using generalized Bayes' theorem with an $\alpha$-posterior across different levels of Gaussian smoothing. Experiments on RF sources demonstrate superior separation performance, with gains of up to $0.95$ in terms of both BER and MSE over classical and existing score-based source separation methods such as BASIS by Jayaram and Thickstun (ICML 2020).

**Strengths:**

Using deep learning for source separation problems has appeared in much research literature. The key contribution of this paper is to provide a new separation method that works well for a superimposition of two discrete sources which produces a joint distribution with multiple equiprobable modes.


This work can be considered a novel combination of well-known techniques.

**Weaknesses:**

The main weak point of this work is a lack of theoretical analysis or intuitions behind the superior separation performance of the proposed method.

**Questions:**

+ In your plots in Fig. 3, $\alpha$-RGS outperforms classical methods such that MF only or LMMSE +MF. Why can this fact hold?
+ Why can your method outperform BASIS for equiprobable multimodal distributions?

**Limitations:**

This is a theoretical work. The authors already finished the checklist as well as stated the limitations of theorems and results.

---

> ### Author Rebuttal · Authors · 2023-08-08
>
> We thank the reviewer for their comments regarding our technical contributions. Below we address the raised questions and concerns.
>
> **Regarding the reviewer’s comments under “weaknesses”:**
> 1. **Theory:** Due to space considerations, we included our theoretical results in the supplementary material. *However, we agree that it is beneficial to readers if we include the main parts  of the developed theory in the manuscript itself. We will move the most important parts of the theory from Appendix B to Section 4 in the final manuscript.* Currently, Section 4 introduces the asymptotic loss function that our algorithm minimizes in the limit of a large number of iterations. Appendix B builds on this, “dissects” equation (14) further, and analytically proves the “mode-seeking” nature of our algorithm for multivariate normal sources, digital RF signals (discrete signals) and Gaussian mixture sources. For each case, we analytically solve for the individual terms in (14) and show that the extrema corresponds to the modes of the underlying source distribution. For the original MAP formulation (6b), we know that the solution(s) correspond to points of maximum probability, i.e., the modes. Hence, the solution of our algorithm, which optimizes (14), approaches the solution of (6b).
>
> **Regarding the questions:**
> 1. **MF and LMMSE:** We recall that MF is the optimal separation solution only when the interference $b$ is Gaussian (see Appendix E.1). Thus, when $b$ is not Gaussian, MF is simply suboptimal and our algorithm outperforms it. The LMMSE estimator is a ***linear*** estimator that minimizes the MSE between the estimated  and true signal. The LMMSE solution is only optimal when both sources are Gaussian. On the other hand, the learned diffusion-based priors model ***non-linearities*** (i.e., it generally extends beyond linear operators) in the data and hence can leverage these during separation. For more details on MF and LMMSE solutions please refer to Appendix E.
> 2. **BASIS:** Please note that Appendix F of the supplementary contains a detailed explanation with the reasons for our model’s superior performance over BASIS. Having carefully reexamined the content of this appendix, we believe that it contains all the necessary clarifications regarding this point. Note that we also provide details about the implementation of BASIS in our experiments. In addition to the above, *we will incorporate some of this intuition into the final version of the manuscript.*

---

> > ### Comment · Reviewer_D4rU · 2023-08-16
> > **Reply to authors' rebuttal**
> >
> > Thank you very much for your rebuttal. The answers are quite satisfactory. Hence, I raise my score.

---

### Official Review · Reviewer_aSGC · 2023-07-05

**Soundness:** 3 good
**Presentation:** 3 good
**Contribution:** 3 good
**Rating:** 6
**Confidence:** 3

**Summary:**

This paper focuses on the problem of single-channel source separation (SCSS) for RF signals with discrete nature. The authors propose to solve the SCSS problem using MAP and use pre-trained diffusion models to approximate the scores for both the source signal and inference signal. To avoid being stuck in a local minimum, the proposed method uses an $\alpha$-posterior and optimizes across multiple noise levels. Experiments show that the proposed method significantly outperforms the baselines (both traditional and learned methods) in both MSE and BER.

**Strengths:**

1) This paper is well-written and it is easy to follow. Given that it is the first work that explores score-based models in the SCSS problem of RF signals and it achieves significant improvement, this work could definitely point out a new direction in this domain and potentially influence other researchers.
2) The idea of using $\alpha$-posterior and randomizing across multiple noise levels could be potentially useful for other score-based optimization problems not limited to the scope of SCSS in RF signals.


**Weaknesses:**

1) The proposed method requires to know the scale factor $\kappa$ (or SIR) and the distribution of the interference signal. However, the scaling factor is not available in real-world scenarios. Also, the interference signals could come from different sources such as WiFi, and Bluetooth. As a result, it may not be possible to train a diffusion model that models all kinds of interference signals in the real world.
2) I think this paper lacks theoretical analysis about: 1) how the proposed algorithm 1 could lead to the local extremum of equation (14), and 2) how minimizing the approximated loss in (9) and (14) could lead to the solution of the original MAP (6b)
3) It would be nice to have more ablation studies about: 1) optimizing across random noise levels vs fixed noise levels, 2) results with and without the zero mean noise in (12)


**Questions:**

1) Could the authors describe more about the suitable conditions in line 205, page 6?
2) In Figure 3, seems like when the SIR is large (around -5dB), the trained SOI model could even outperform the analytical one. Could the authors explain more about that?


**Limitations:**

The authors have adequately addressed the limitations

---

> ### Author Rebuttal · Authors · 2023-08-08
>
> We thank the reviewer for the feedback and helpful questions. We shall now address the raised questions and concerns.
>
> **Regarding the reviewer’s comments under “weaknesses”:**
> 1. **Knowledge about the scaling factor**: *This is an excellent point and we hope to address it in future work by introducing a novel extension to our algorithm that jointly optimizes over the range of possible scaling factors as well.* Many communication systems have power constraints and equalization capabilities, and with the endowment of such knowledge it is possible to estimate the signal to interference ratio (SIR) within reasonable margin. Equalization/normalization of the SOI’s power can be performed, e.g., by leveraging header information (metadata) and the SIR can be inferred from the mixture with the knowledge of the former. Thus, as a first step towards this goal, we assume knowledge of $\kappa$ in this work.
> 2. **Interference signals from different sources:**  *Developing a library of priors in a cost-effective manner will be a focus of our future work.* It is true that the wireless ecosystem is growing at a rapid pace. We envision a system of plug-and-play priors where the receiver is given access to an ML backbone with diffusion-based priors, ***each*** trained on a different signal type, e.g., Bluetooth or WiFi. During transmission, a simple detector might detect the presence of a Bluetooth interference signal and the receiver can plug in the learned prior for Bluetooth signals to recover the signal-of-interest. We believe that such technology is vital for future communication standards such as 6G and we will focus our future efforts on researching new ways to efficiently learn these priors. For details on our diffusion model training setup, please see Appendix D.
> 3. **Theoretical results:** Due to space considerations, we included our theoretical results in the supplementary material. *However, we agree that it is beneficial to readers to include some of the developed theory in the manuscript itself. We will move the most important parts of this theory from Appendix B to Section 4 in the final manuscript.*
> 4. **Connection between formulations:** Section 4 introduces the asymptotic loss function that our algorithm minimizes in the limit of a large number of iterations. Appendix B builds on this, “dissects” equation (14) further, and analytically proves the “mode-seeking” nature of our algorithm for multivariate normal sources, digital RF signals (discrete signals) and Gaussian mixture sources. For each of these cases, we analytically solve for the individual terms in (14) and show that the extrema corresponds to the modes of the underlying source distribution. For the original MAP formulation (6b), we know that the solution(s) correspond to points of maximum probability, i.e., the modes. Hence, the solution of our algorithm, which optimizes (14), approaches the solution of (6b).
> 5. **Ablation studies:** We decided to use all noise levels in our algorithm since diffusion models capture statistical structures at different resolutions based on the amount of noise added in the forward process. This is pictorially depicted in the middle and rightmost plots of Figure 1, where the distribution is smoothened and less peaky with increasing amounts of Gaussian smoothing.  We emphasize that the BASIS algorithm also uses all noise levels but the noise is gradually annealed over time via a ***fixed*** schedule. Each outer iteration of BASIS essentially tries to separate the components using a fixed noise level. As mentioned in the manuscript and in Appendix F, tuning this schedule is difficult, and more importantly still leads to underperformance in comparison to our method. As for the second ablation study without the subtractive noise, initial experiments demonstrated benefits with this additional term, but we will re-run the experiment for a few cases to be included in the supplementary material. We thank the reviewer for this suggestion.
>
> **Regarding the questions:**
> 1. **Suitable conditions:** While it is analytically challenging to answer the necessary conditions under which this holds, we are able to provide sufficient conditions for separability of the sources under equation (14): If the two sources, $s$ and $b$,  are discrete and the super constellation (see Appendix C.1 in the supplementary)  is uniquely decodable, i.e., the mapping between the symbols in the super constellations of each source is unique, perfect recovery is possible. However, in this work we intentionally focus on general signal types and mixtures, where perfect separability is not necessarily guaranteed and performance bounds are not analytically tractable. Providing theoretical guarantees in more complicated scenarios is a topic of broad interest and we hope to address some of these problems in future work.
> 2. **Trained model vs.analytical model:**  We attribute this degradation to numerical instabilities during the computation of the analytical score in the symbol domain (before pulse-shaping, i.e., $a$ where $s = Ha$). Computing the score of the pulse-shaped symbols ($s = Ha$) is extremely challenging. For more details on the digital communication pipeline please refer to Appendix C. Appendix G ((G.1) – (G.4)), details the calculation of the analytical score for the QPSK SOI in our experiments. As shown in (G.4), we make an approximation via the pseudo-inverse of $H$. At low interference levels (high SIR), this approximation can lead to small errors in symbol recovery, due to the negligible additive noise. On the other hand, the trained model directly learns the score of $s = Ha$ and thus circumvents such approximation errors.

---

> > ### Comment · Reviewer_aSGC · 2023-08-12
> > **About the Rebuttal**
> >
> > I thank the authors for the detailed response. I think the majority of my concerns have been addressed.

---

### Official Review · Reviewer_J15P · 2023-07-06

**Soundness:** 3 good
**Presentation:** 3 good
**Contribution:** 4 excellent
**Rating:** 6
**Confidence:** 4

**Summary:**

A new method for separating superimposed sources using diffusion-based generative models is proposed (alpha-RGS).
The method relies on separately trained statistical priors of independent sources and is guided by maximum a posteriori estimation with an α-posterior.
Experimental results with RF mixtures demonstrate that the method results in a BER reduction of 95% over classical and existing learning-based methods.



**Strengths:**

The authors propose a new method for source separation called α-RGS (α-posterior with Randomized Gaussian Smoothing).
The method uses the (approximate) score from pre-trained diffusion models to extend maximum a posteriori (MAP) estimation using generalized Bayes' theorem with an α-posterior.
α-RGS outperforms classical signal processing and annealed Langevin-dynamics-based approaches for RF source separation.

**Weaknesses:**

.

**Questions:**

1. It would be interesting to see how the model performs in different channel conditions, with different types of noise other than additive white Gaussian noise (AWGN). This would give us a better understanding of the model's robustness to different noise conditions.

2. The BER curve of the proposed model is noisy because it was generated using a small number of test examples. This noise will likely disappear if we sample more test examples, which will allow us to compute the exact improvement (in terms of dB) of the proposed model over the baseline.

**Limitations:**

.

---

> ### Author Rebuttal · Authors · 2023-08-08
>
> We thank the reviewer for the feedback and helpful questions. We shall now address the raised questions and concerns.
>
> **Regarding the questions:**
> 1. **Statistics of noise:** Thank you for the question. We would first like to clarify that in a typical communication setup with a transmitter, channel and receiver, it is often assumed that the channel noise is AWGN. Mathematically, the output $y$ is related to the input $s$ as $y = s + w$ where $w$ is AWGN. In our source separation setup we consider an interference channel $y = s + b$ where $b$ is no longer constrained to be AWGN and thus departing the channel noise model. The generality of our approach is rooted in learning the underlying structures of the interference, which can often be far more complicated than AWGN.
> 2. **BER curves:** *We will increase the size of the test set for generating the curves for the final version of the manuscript as much as possible.*

---

> > ### Comment · Reviewer_J15P · 2023-08-18
> >
> > Thank you for address my concerns.

---

### Official Review · Reviewer_jCQT · 2023-07-10

**Soundness:** 2 fair
**Presentation:** 3 good
**Contribution:** 3 good
**Rating:** 5
**Confidence:** 3

**Summary:**

This paper investigates a novel Bayesian approach for separating the superposition of two sources based on diffusion generative models. The problem is motivated by application to the spectrum of radio-frequency (RF) communication systems. Several experiments using real datasets on RF  mixtures demonstrate that the proposed method reduces the bit error rate, which is the stared measure of performance for this application.

**Strengths:**

The strengths of this paper are as follows:

+ It investigates a well-known problem, i.e., source separation, by using modern tools, i.e.,  diffusion generative models. In particular, the use of diffusion generative models to model the prior in the Bayesian source-separation framework is novel and appealing.
+ Several numerical results are provided by using real datasets.
+ The paper is well-written and the contribution is clearly stated.

**Weaknesses:**

The weaknesses of this paper are as follows:

+ The paper introduces the problem of multi-source separation. However, it focus on the case of two sources and the methodology specializes very much on the presence of only two sources.
+ There is no theory or more specifically, there are no guarantees on performance of the proposed method to operate the underlaying sources. Notice that a lot have been done in this area and it would important to understand the limitations of the proposed method or at least to derive sufficient condition to ensure that the sources can be separated.
+ The applications to future wireless communication systems are interested but very much limited. In particular, those are applications are not the main focus of the ML community. it would have been beneficial to show the proposed can be used to other type of data since they are many practical cases (e.g. sounds and speech) for which source separation is requested.

**Questions:**

+ The paper lacks of theoretical results showing the limitation of the proposed method and the assumptions on the data distribution to be able to perform source separation.
+ I strongly suggest the authors to further investigate other scenarios for which source separation is requested which could be of major interested for the ML community.

**Limitations:**

The limitation are not well discussed; In particular, it is not clear the underlaying assumptions necessary to  satisfy the source separation.

---

> ### Author Rebuttal · Authors · 2023-08-08
>
> We thank the reviewer for taking the time to understand the technical contributions in our work. Below we shall address the raised questions and concerns.
>
> **Regarding the reviewer’s comments under “weaknesses”:**
> 1. **Multi-source Separation:** To make the exposition clearer, we focus on the two-component case as mentioned in the beginning of our manuscript on line 3. We next explain how this readily defines a procedure for multi-source separation as well. Consider a three-component mixture $y = s + n + w$. This can be separated using our algorithm in two different ways — a) by treating the interference as $b = n + w$ in the first pass of our algorithm, we can obtain $\hat{s}$, and then separating $\hat{b} = y - \hat{s}$ in a second pass we obtain $\hat{n}$ and $\hat{w}$; b) by using the three priors simultaneously to obtain estimates $\hat{s}$, $\hat{n}$ and using the constraint $y = s + n + w$. In terms of applications, a very important scenario in RF is the interference rejection scenario, where we are mostly interested in the recovery of our signal of interest only. In this setup, the interference could potentially describe the background/environment, which in practice could be a superposition of multiple devices. *Recovering the signals in a multi-source setting (particularly in a single-pass) is a subject of future work — and given its relevance in the context of such problems, we will make this clearer in our discussion and concluding remarks.*
> 2. **Theory and performance guarantees:**  The conditions for perfect signal separation can be described for certain simple models. One such sufficient condition is as follows: If the two sources, $s$ and $b$,  are discrete and the super constellation (see Appendix C.1 in the supplementary)  is uniquely decodable, i.e., the mapping between the bit representation and symbol representation of the signals is unique, perfect recovery is possible. However, in this work we are focusing on general signal types and mixtures, where perfect separability is not necessarily guaranteed and performance bounds are not analytically tractable. A more involved characterization of our algorithm’s convergence to the modes beyond Section 4 can be found in Appendix B. *We agree that understanding the sufficient conditions and easy access to the theory in Appendix B are important. We will modify Section 4 of the final manuscript to include the sufficient condition above along with a condensed version of the  analysis from Appendix B.*
> 3. **Relevance to ML community:** While we agree that image and audio separation are important areas of research, we found existing score-based source separation methods initially developed for these domains to underperform on RF data (e.g., BASIS). Moreover, the underlying discreteness of typical RF signals distributions present new challenges to the ML community that generally do not arise with image or audio data. Recently, there has been tremendous interest in using ML techniques for digital communications (please see some sample publications below) and hence we strongly believe that such new application areas are of interest to the broad ML community. There are a plethora of challenging problems in the wireless domain and we believe the ML community can help take significant steps towards better solutions. ML is without doubt a central technology to future wireless standards such as 6G, where the increasing demand of bandwidth combined with high reliability will require sophisticated interference rejection/source separation algorithmic solutions. We hope that this work can help shed additional light on new and rapidly developing areas of ML.
>
> **Regarding the questions and limitations:**
> 1. **Sufficient conditions:** Thank you for pointing this out as this may have not been very clear from our manuscript. Please see the response above under “Theory and performance guarantees” for the sufficient condition. *We will include this condition in Section 4 of the final manuscript.*
> 2. **Applications to other modalities:** The paper is intentionally focused on digital communication signals due to the fundamental differences of the signals' statistics, which is essentially a different realm that will (possibly, and perhaps most likely) require (at least) new building block for successfully functioning DNNs (where "success" is in terms of significant gains over classical model-based methods). *However, we agree with the reviewer that it would be interesting to extend our experiments to other modalities as well. We appreciate the suggestion and we will add these experiments to our future work.*
>
> **A few recent publications (apart from the closely related works already cited in the manuscript) that leverage ML for digital communications:**
> 1. **Specific to single-channel source separation of RF/wireless signals with ML:**
> - M.Zhao, et al. Single-channel blind source separation of spatial aliasing signal based on stacked-LSTM.Sensors, 2021.
> - X. Hou and Y. Gao. Single-channel blind separation of co-frequency signals based on convolutional network. Digital Signal Processing, 2022
> - H. Ma, et al. A novel end-to-end deep separation network based on attention mechanism for single channel blind separation in wireless communication. IET Signal Processing, 2023.
> 2. **ML in wireless/RF communications for other problems:**
> - T.  O’Shea, et al. Over-the-air deep learning based radio signal classification.IEEE J.Sel.Topics Signal Process., 2018.
> - T. O’Shea and J. Hoydis. An introduction to deep learning for the physical layer.IEEE Transactions on Cognitive Communications and Networking, 2017.
> - Y. Eldar, et al. Machine Learning and Wireless Communications.Cambridge: Cambridge University Press, 2022.
> 3. **NVIDIA Sionna Toolkit for Next Generation Communications Research (which we use in our experiments):**
> - Hoydis, Jakob, et al."Sionna: An open-source library for next-generation physical layer research." arXiv preprint arXiv:2203.11854 (2022).

---

> > ### Comment · Reviewer_jCQT · 2023-08-18
> >
> > Thank you very much for your rebuttal. The authors have answered my questions quite satisfactory. Accordingly,  I will increase my score.

---

### Author Rebuttal · Authors · 2023-08-09

We thank the reviewers for taking the time to review our work proposing a novel algorithm that leverages diffusion-based priors and randomized levels of Gaussian smoothing with applications source-separation in the RF domain, a relatively new domain area to the ML community.  We appreciate all the comments and questions that have been raised in the reviews, and by this global response we hope to respond to the reviewers collectively so as to address a few concerns.

Regarding the relevance of source separation in the context of wireless signals, we would like to reiterate that the source separation problem within the RF domain comes with a different set of technical challenges as well as academically interesting questions. We believe that such challenges are not only of interest to the ML community, but moreover, ML researchers can help shape novel algorithms for AI-enhanced next-generation communication technology (and indeed this is the vision for 6G).

As for the theoretic aspects of our work, due to space constraints, the development and understanding of our proposed algorithm from first principles was deferred to Appendix B.  Taking into account the reviewers’ comments, and in order to enhance readability, we will move some of the important theoretical results from Appendix B to the manuscript. We also appreciate the immense interest in a deeper understanding of the theory, and while we are able to provide sufficient conditions for separability in simple cases, there are indeed parts of the problem which might  be analytically intractable—e.g.conditions for perfect separability or the lower bound on the performance for general classes of signals—which are intriguing theoretic investigations which we will pursue in future work.

We understand that a detailed explanation of the experimental setup, digital communications terminology, details about the baselines and a detailed overview of the results are beneficial to readers. However, space constraints preclude us from including extensive details in the manuscript beyond what has already been provided, and are therefore included in Appendices C - G. With the additional space provided in the final manuscript, we will add a paragraph that describes the paper and appendix organization to help readers navigate the work efficiently. We will also move additional important details regarding the experimental setup to Section 5 in the final manuscript.

---

> ### Comment · Area_Chair_7jRF · 2023-08-18
> **Thank you for the rebuttal**
>
> Dear authors,
>
> thank you for providing a rebuttal. Some of the reviewers have already replied, so this is just to let you know that I am in contact with the remaining ones as well.
>
> Best,
> Your AC

---

### Decision · Program_Chairs · 2023-09-21

**Decision:**

Accept (poster)

**Comment:**

This paper proposes a new method for source separation based on generative models. The motivation comes from RF systems and experimental results with RF mixtures are provided.

The idea of using generative models for the prior in the Bayesian source-separation framework is interesting and novel, the empirical results are convincing, and the paper is well written. The authors have provided a thoughtful rebuttal which has addressed a number of concerns expressed by the reviewers, who have now reached a consensus towards accepting the paper. Based on my own reading of the reviews, rebuttal and paper, I agree with this view and I am happy to recommend acceptance.